# Integrated Omic Approaches Reveal Molecular Mechanisms of Tolerance during Soybean and *Meloidogyne incognita* Interactions

**DOI:** 10.3390/plants11202744

**Published:** 2022-10-17

**Authors:** Fabricio B. M. Arraes, Daniel D. N. Vasquez, Muhammed Tahir, Daniele H. Pinheiro, Muhammed Faheem, Nayara S. Freitas-Alves, Clídia E. Moreira-Pinto, Valdeir J. V. Moreira, Bruno Paes-de-Melo, Maria E. Lisei-de-Sa, Carolina V. Morgante, Ana P. Z. Mota, Isabela T. Lourenço-Tessutti, Roberto C. Togawa, Priscila Grynberg, Rodrigo R. Fragoso, Janice de Almeida-Engler, Martin R. Larsen, Maria F. Grossi-de-Sa

**Affiliations:** 1Postgraduate Program in Cellular and Molecular Biology (PPGBCM), Federal University of Rio Grande do Sul (UFRGS), Porto Alegre 91501-970, RS, Brazil; 2Embrapa Genetic Resources and Biotechnology, Plant-Pest Molecular Interaction Laboratory (LIMPP) and Bioinformatics Laboratory, Brasilia 70770-917, DF, Brazil; 3National Institute of Science and Technology (INCT PlantStress Biotech), Brasilia 70770-917, DF, Brazil; 4Postgraduate Program in Genomic Sciences and Biotechnology (PPGCGB), Catholic University of Brasilia (UCB), Brasilia 71966-700, DF, Brazil; 5Department of Biochemistry and Molecular Biology, University of Southern Denmark, 5230 Odense, Denmark; 6Department of Biological Sciences, National University of Medical Sciences, The Mall, Rawalpindi 46000, Punjab, Pakistan; 7Postgraduate Program in Bioprocess Engineering and Biotechnology (PPGEBB), Federal University of Paraná (UFPR), Curitiba 80060-000, PR, Brazil; 8Postgraduate Program in Molecular Biology (PPGBiomol), University of Brasilia (UnB), Brasília 70910-900, DF, Brazil; 9Minas Gerais Agricultural Research Company (EPAMIG), Uberaba 31170-495, MG, Brazil; 10Embrapa Semiarid, Petrolina 56302-970, PE, Brazil; 11INRAE, Université Côte d’Azur, CNRS, Institut Sophia Agrobiotech, 06903 Sophia-Antipolis, France; 12Embrapa Agroenergy, Brasilia 70770-901, DF, Brazil

**Keywords:** root-knot nematode, transcriptome, proteome, differential expression, phenylpropanoids

## Abstract

The root-knot nematode (RKN), *Meloidogyne incognita*, is a devastating soybean pathogen worldwide. The use of resistant cultivars is the most effective method to prevent economic losses caused by RKNs. To elucidate the mechanisms involved in resistance to RKN, we determined the proteome and transcriptome profiles from roots of susceptible (BRS133) and highly tolerant (PI 595099) *Glycine max* genotypes 4, 12, and 30 days after RKN infestation. After in silico analysis, we described major defense molecules and mechanisms considered constitutive responses to nematode infestation, such as mTOR, PI3K-Akt, relaxin, and thermogenesis. The integrated data allowed us to identify protein families and metabolic pathways exclusively regulated in tolerant soybean genotypes. Among them, we highlighted the phenylpropanoid pathway as an early, robust, and systemic defense process capable of controlling *M. incognita* reproduction. Associated with this metabolic pathway, 29 differentially expressed genes encoding 11 different enzymes were identified, mainly from the flavonoid and derivative pathways. Based on differential expression in transcriptomic and proteomic data, as well as in the expression profile by RT–qPCR, and previous studies, we selected and overexpressed the *GmPR10* gene in transgenic tobacco to assess its protective effect against *M. incognita*. Transgenic plants of the T_2_ generation showed up to 58% reduction in the *M. incognita* reproduction factor. Finally, data suggest that *GmPR10* overexpression can be effective against the plant parasitic nematode *M. incognita*, but its mechanism of action remains unclear. These findings will help develop new engineered soybean genotypes with higher performance in response to RKN infections.

## 1. Introduction

Soybean [*Glycine max* (L.) Merryll] is one of the most economically important crops and is widely used as a source of protein and oil for human food, animal feed, and biofuel production [1,2,3]. Currently, Brazil is the largest soybean producer, accounting for more than 35% of global soybean production, followed by the United States and Argentina [4]. Root-knot nematodes (RKNs; *Meloidogyne* spp.) are highly relevant plant parasites that infect a broad range of crops worldwide [5,6]. Among these nematodes, the southern root-knot nematode (*Meloidogyne incognita)* seriously compromises soybean crop yields and represents a major problem in soybean-producing regions [7]. After hatching, the infective second-stage juveniles (J2s) of these endoparasitic nematodes move through the soil up to the root tissues of the host plant. Then, they establish a specialized feeding site, which consists of differentiated root cells (giant cells) that provide water and nutrients required for nematode development and reproduction during their life cycle [8,9,10]. The parasitism process begins with the insertion of a stylet structure into the plant cells and the subsequent secretion of effector molecules from the esophageal gland cells, which triggers morphological and physiological changes in the infected plant cells, leading to the formation of giant cells and galls [11]. The galls cause interference in the upward translocation of nutrients and water from the infected roots to the leaves and developing seeds. As a result, the growth and yields of soybean are heavily reduced [12]. In addition, the nutritional deficiency caused by *M. incognita* infection usually induces foliar chlorosis, which in turn decreases the photosynthetic rate and chlorophyll content in soybean plants [13,14].

Due to the great importance of plant parasitic nematodes to world agriculture, studies that aim to understand the mechanisms of immune responses of several host plants to this pathogen have been developed. In general, the zig zag model describes two main signaling pathways that can be modulated primarily in host plants, pattern-triggered immunity (PTI) and effector-triggered immunity (ETI) [15]. After recognition of specific molecules of the pathogen (in the case of RKNs, nematode-associated molecular patterns, NAMPs) by cell surface-localized pattern recognition receptors (PRRs), the first layer of defense, called PTI, is activated [15,16]. In this first line of defense, the same metabolic pathways are usually modulated, which include callose deposition, accumulation of calcium in the cytoplasm, activation of mitogen-activated protein kinase (MAPKs), mainly MAPK3-6, production of reactive oxygen species (ROS), which induces the expression of several defense-related genes [17]. To date, the main NAMPs recognized by host plants belong to the family of ascarosides, which are small lipid molecules that act as pheromones in nematodes and control a variety of sex-specific and social behavior [18]. Among the seven main ascarosides described in plant nematodes, ascr#18 has already been identified in *M. incognita*, *M. hapla*, and *M. javanica*, as well as in cyst and lesion nematodes [19,20]. Even though several theoretical and experimental evidences indicate ascarosides as important NAMPs, these studies still need to be complemented since no receptor for ascarosides has been identified in plants [21]. To date, few PRRs have been identified in response to plant nematodes: *AtNILR* and the co-receptor *AtBAK1*, identified in *Arabidopsis thaliana* [22,23], as well as the orthologs of *BAK1* in tomato (*SlSERK3A* or *SlSERK3B*) [24]. On the other hand, the second layer of defense called ETI, the recognition of nematode effectors introduced into the host plant during infection is mediated by resistance (R) proteins [20,25]. This immune response is often, but not exclusively, visualized by the hypersensitivity response (HR), which is characterized by a rapid cell death at the point of pathogen entry and reduction of the nematode’s feeding site [26]. Several nematode effectors [27,28,29,30,31,32,33] and plant *R* genes [34,35,36,37,38,39] have been characterized. Studies have even showed evidence in soybean the participation of non-canonical *R* genes in resistance to *M. incognita* [40].

Currently, *M. incognita* control is mainly achieved through the use of tolerant soybean genotypes, crop rotation with non-host plants, and nematicides. Soybean breeding programs have allowed the introgression of resistance genes into soybean varieties with attractive agronomic traits. This is achieved by performing crosses with genotypes that have natural genetic resistance to nematodes; however, this approach can be time-consuming and laborious due to the complexity of plant disease resistance mechanisms [41,42,43]. Moreover, the selection of virulent nematode populations has been reported for most known resistance sources [44]. As an alternative, the integration of omics can potentially uncover new sources of plant resistance to *M.*
*incognita* for exploration either through conventional cross-breeding programs or plant genome engineering tools, such as transgenic breeding and CRISPR technologies [45,46].

Numerous soybean lines with variable levels of resistance to nematodes have been identified from soybean germplasm collections and used in breeding programs. The introgression of the PI 595099 soybean genotype with a high level of resistance to *M. incognita* was developed by the Georgia Agricultural Experiment Station (EUA), while BRS 133, by contrast, is a susceptible genotype developed by the National Soybean Research Center—Embrapa Soybean (Brazil). Both genotypes are agriculturally important and have been used in soybean breeding programs and in studies to identify genetic resistance to parasitic nematodes [47,48,49,50,51,52,53]. Recently, an increasing number of studies exploring the mechanisms underlying the resistance and susceptibility of diverse crops to *M.*
*incognita* have been performed using omic approaches [51,54,55,56,57,58,59,60,61,62,63,64,65,66,67]. Regardless, little is known about the response of PI 595099 and BRS 133 genotypes to *M. incognita* infection at the molecular level. Therefore, a comparison of the molecular response induced and/or repressed during host–nematode interaction in these contrasting genotypes can provide insights into the basis of soybean response to *M. incognita*. Furthermore, expanding our understanding of how soybean responds to *M. incognita* infection using molecular and genomic tools will help to apply biotechnological approaches to develop soybean varieties with increased resistance to *M. incognita*, as well as reduce the losses in crop production caused by this devastating nematode.

In the present study, we performed an integrated multi-omics analysis of highly tolerant and susceptible soybean genotypes in response to *M. incognita* parasitism. The transcriptomic and proteomic analyses performed in our study will help improve the knowledge of the molecular mechanism of soybean-*M. incognita* interaction and unravel candidate genes associated with tolerance/resistance and susceptibility to this plant nematode that can be used as the basis for genetic soybean breeding.

## 2. Results

### 2.1. Morphology and Fitness Evaluation of Infected Soybean Genotypes

From the best contrasting genotypes for infection with *M.*
*incognita*, the BRS 133 was selected as the susceptible genotype, and the genetic accession PI 595099 was pre-selected as highly tolerant to *M. incognita*. To date, no genotypes or soybean genetic accessions have been described as resistant/immune to parasitism by RKNs; for this reason, a highly tolerant accession was selected. Previous studies have tested the response of genotypes BRS 133 and PI 595099 to the infection with *M. javanica*; however, this is the first report that evaluates the response of these two genotypes to *M. incognita*. Thus, 60 days after inoculation (DAI; approximately two life cycles of *M. incognita*), the reproductive factor (RF) of the nematode in both genotypes was determined. As expected, the susceptible (BRS 133) and high-tolerant (PI 595099) phenotypes were confirmed, with RF values of 51.3 ± 1.2 and 2.7 ± 0.7, respectively (Figure 1).

In a second moment, morphological analyses were carried out to evaluate the dynamics of infection with *M. incognita* in both genotypes. First, the determination of penetration efficiency (PEf) was an essential analysis to obtain indications of possible initial mechanisms of susceptibility/tolerance, such as physical barriers, which prevent the entry of the nematode and subsequent infection of the root tissue. BRS 133 and PI 595099 genotypes had remarkably similar PEf values (approximately 73–75%) (Figure 1a,f). These data provide evidence that the penetration by *M.*
*incognita* in the root tissue in both genotypes would not be an influencing factor in the observed RF values. Furthermore, this data suggests that the mechanisms activated against *M. incognita* in the two soybean genotypes are triggered after the establishment of the nematode feeding site through molecular regulation. In addition, it was observed that *M. incognita* J2s preferred young secondary roots, with entry generally occurring in large numbers of nematodes per entry site (5–16 J2s per root tip) (Figure 1a,f).

In addition to the penetration coefficient, the progression of infection in both genotypes during one *M. incognita* life cycle was evaluated. Thus, samples were selected at 4 DAI (number of days that, according to the literature, *M. incognita* can survive in the soil with only its energy reserves) and 30 DAI (approximate end of the first *M. incognita* life cycle in soybean roots). In addition to these two conditions, 12 DAI was selected because, after this time, the first galls appeared. In this time course, the morphological analyses performed with microscopic sections of collected tissues demonstrated a standard infection cycle from the establishment of the feeding site (post-parasitic J2) to the formation of egg masses external to the galls (female). It is important to highlight the distinct morphology between the galls of the two genotypes, with most galls observed in the genotype BRS 133 being elongated (Figure 1b), whereas the same structure observed in the genotype PI 595099 is globular, as an appendix to the root (Figure 1g). According to the micrographs, the galls observed at 12 and 30 DAI in both genotypes had characteristic structures composed by hypertrophied multinucleated giant cells generated by multiple mitotic division of vascular root cells without cytokinesis (Figure 1d,e,i,j).

In addition, hyperplasia of the cells around the giant cell, characterizing the gall structure, was evident under all conditions analyzed. Thus, few differences were observed between the morphology of the galls of susceptible (BRS 133) and highly tolerant (PI 595099) genotypes; only an apparent delay in the development of giant cells in the susceptible genotype was highlighted, which may be an indication of more efficient feeding site establishment.

### 2.2. Transcriptome and Proteome General Analysis

In total, 959.4 million paired-end (PE) reads were sequenced from twenty-four cDNA libraries of two soybean genotypes infected with *M. incognita*. The average library size was 40.0 million PE reads (Table 1; Appendix A). The 726.1 million PE high-quality filtered reads (1.5 billion reads) were pseudoaligned into the soybean transcriptome predicted from *G. max* Williams 82 genome assembly (Wm82.a2.v1) and further used for digital expression analysis. The raw data were uploaded to the NCBI SRA database under bioproject number PRJNA750661.

In parallel, the proteome results of twenty-four samples of total protein of the same two genotypes evaluated in the transcriptomics experiments were also analyzed. Based on the predicted proteins in the Wm82.a2.v1 soybean genome assembly, more than 11,400 protein groups were identified in all conditions evaluated per genotype (11,462 protein groups in roots of the BRS 133 genotype and 11,488 in the PI 595099 genotype) (Table 2; Appendix A). Similar to what was done for the transcriptome data, the raw proteome data were uploaded into a specific database (ProteomeXchange) under the ID PXD028483.

### 2.3. Overview of Differential Expressed Genes (DEGs) and Differential Expressed Proteins (DEPs)

A total of 5842 DEGs (5317 unique) and 469 DEPs (400 unique) were statistically significant (false discovery rate, *FDR* < 0.05; log_2_ (fold change, FC) ≥2 or ≤−2) in the BRS 133 genotype, while PI 595099 presented 7041 DEGs (6636 unique) and 720 DEPs (564 unique) (*FDR* < 0.05; log_2_(FC) ≥1 or ≤−1) during the 30 DAI with *M. incognita*. A more detailed view shows that DEPs shared in both genotypes ranged from 22% to 39% (Appendix A), and a larger representation of DEPs was observed in the PI 595099 genotype at all evaluated time points (41–68%). Similar trends were noted in the transcriptomics analysis (Appendix A), for which 17% to 40% of the DEGs were identical for both genotypes. Furthermore, most DEGs were exclusively found in PI 595099 at 12 and 30 DAI (50% and 42%, respectively). In contrast, at 4 DAI, the highest proportion (74%) of DEGs was unique for genotype BRS 133.

As expected, regarding the effect of time on the number of DEGs and DEPs, a similar tendency was observed for both genotypes. A remarkable proportion of both DEGs (~84%) and DEPs (~76%) were found only after 30 days of infection (Appendix A). Otherwise, in BRS 133, less than 4% of DEGs and DEPs were common all-time points evaluated, and in PI 595099, the number was even lower (<1%). In addition, at 4 DAI, 3–4 DEGs were exclusively significant, whereas DEPs ranged from 7% to 16%. In addition, 95 DEGs were identified as presenting the opposite expression level between both genotypes at least once (Appendix A).

Last, a direct comparison between DEGs and predicted genes encoding DEPs revealed 162 genes shared between the transcriptome and proteome of soybean infected with *M. incognita* (Appendix A). Moreover, for 35 of these genes, the expression level (over- or underexpressed) was opposite between the two approaches, while for the remaining 127 genes, it was the same. In addition, of the 162 genes, 37 (23%) were observed in both genotypes, whereas 81 (50%) and 44 (27%) were unique to PI 595099 and BRS 133, respectively.

### 2.4. Functional Enrichment of Gene Ontology (GO) Terms

Overall, 66 and 26 GO terms were enriched in the soybean transcriptome (Figure 2) and proteome (Appendix A), respectively. Some major processes and functions, including carbohydrate metabolism, catalytic activity, oxidation-reduction, responses to stimulus, signal transduction, regulation of GTPase, lipid metabolism, enzyme regulation, and binding, were enriched in both datasets.

The 30 DAI condition showed the highest number of enriched GO terms. Regarding the transcriptome, at this timepoint, both genotypes showed underexpressed genes mainly related to catalytic processes, glucan metabolism, and cell wall organization. However, PI 595099 exhibited some GO terms that were not enriched in BRS 133, including terms related to mitotic nuclear division, lipid metabolism, and developmental processes.

On the other hand, overexpressed genes were linked to photosynthesis, signal transduction, and some oxidoreduction processes in both genotypes. Interestingly, transport processes, phototropism, and cytokinin metabolism were exclusively enriched in PI 595099. In the proteome, overexpressed genes in PI 595099 were associated with the metabolism of molecules, such as asparagine, sucrose, and glycogen, whereas in BRS 133, they were related to the inhibition of endopeptidases. Finally, a set of underexpressed genes related to the metabolism of methionine and transferases were enriched in PI 595099.

In contrast to the plants evaluated at 30 DAI, the remaining treatments showed a lower number of associated GOs (transcriptome, 9 at 12 DAI and 16 at 4 DAI; proteome, 12 at 12 DAI and 6 at 4 DAI). Concerning the transcriptome, 12 DAI genotype PI 595099 exhibited overexpression of genes mainly related to DNA replication, transcription, and cell proliferation. Additionally, for the same treatment, the proteome showed protein overexpression in proteolysis and peroxidation in PI 595099. Finally, 4 days after inoculation, most overexpressed genes were particularly enriched in BRS 133 (e.g., embryo development, transcription, and chitin-binding). A similar pattern was observed for overexpressed proteins, which were mostly linked to brassinosteroid signaling pathways.

### 2.5. Enrichment of Metabolic Pathways

In total, 25 and 3 pathways were enriched in the DEG and DEP sets, respectively (Figure 3). Of these, only retinol metabolism was not enriched in the samples at 30 DAI. Furthermore, 16 pathways were associated with the downregulation of transcripts or proteins in the mentioned treatment. Remarkably, overexpressed pathways were enriched solely in the genotype PI 595099. Additionally, three and one pathways were enriched in the treatments at 12 DAI and 4 DAI, respectively. Only phenylpropanoid biosynthesis was enriched in all treatments (Figure 3). Finally, it is important to highlight that some signaling pathways, such as mTOR, PI3K-Akt, relaxin, and thermogenesis, were enriched in both overexpressed and underexpressed DEGs.

### 2.6. Phenylpropanoid Metabolism in Soybean-M. incognita Interaction

The analysis of metabolic pathway enrichment suggests some differences in plant defense-associated mechanisms that reinforce the contrasting phenotypes of the soybean genotypes BRS 133 and PI 595099 infected with RKN. Overall, proteomic data show suppression of enzymes from phenylpropanoid biosynthesis in the susceptible genotype BRS 133 (Figure 3 and Figure 4), with a lower abundance of differentially expressed proteins from the early to the late stages of nematode infection (4 DAI and 30 DAI). Interestingly, these proteins were more highly expressed in the tolerant genotype PI 595099 at the early-to-intermediate and late stages of pathogenesis (12 DAI and 30 DAI, respectively), indicating a fine-tuned and integrated regulation of pathogen-triggered cell signaling and secondary metabolism. We investigated the abundance of genes and proteins from phenylpropanoid biosynthesis and other metabolite-related pathways differentially expressed in the susceptible and tolerant genotypes (Table 3).

Our analysis found 29 differentially expressed genes encoding 11 different enzymes involved in phenylpropanoid metabolism, mainly from the flavonoid and derivative pathways (Table 3). In the tolerant genotype, the genes 4-coumarate-CoA ligase (*4CL*), caffeoyl-CoA 3-o-methyltransferase (*CCoAOMT*), isoflavone 4-o-methyltransferase (*IOMT*), leucoanthocyanidin dioxygenase (*LDOX*) and laccase (*LCC*) were overexpressed during the early stages of pathogenesis (up to 12 DAI), suggesting mainly early activation of defensive pathways in tolerant soybean genotype. Only cinnamate-4-hydroxylase (*C4H*), flavonoid 3′-hydroxylase (F3′H), and flavonoid 6′-hydroxylase (F6′H) were overexpressed at 30 DAI. Otherwise, most genes were underexpressed in the late stages of nematode infection in the tolerant genotype, especially those encoding other paralogs of LCC enzyme and Cinnamyl-Alcohol Dehydrogenase (CAD) (Table 3). In contrast, the genes encoding the flavonoid 3′-hydroxylase (F3′H) and LCC enzymes were underexpressed in the early stages of pathogenesis in the susceptible genotype and remained underexpressed up to 30 DAI, whereas only two genes, encoding LDOX and 55 cinnamate-4-hydroxylase (C4H) enzymes, were overexpressed.

### 2.7. Transcription Factor Families

Among the DEGs, 460 were annotated as encoding transcription factors (TFs) belonging to 15 families, including basic leucine zipper domain (bZIP), GATA family (GATA), GRAS-domain (GRAS), Heat shock factors (HSFs), DNA-binding MADS domain (MADS-box), KNOX/ELK homeobox (KNOX/ELK), DNA-binding domain MYB domain (MYBs), plant AT-rich sequence- and zinc-binding (PLATZ), DNA-binding domain WRKY (WRKY), basic helix-loop-helix TFs (bHLH), homeobox protein BEL1 (BEL1), c2h2/c2hc zinc fingers (C2H2/C2HC ZF), apetala 2 (AP2), DNA-binding domain NAC domain (NAC) and growth regulating factors (GRFs). (Figure 5). Our data show that during soybean root inoculated with *M. incognita*, the expression of TFs was highly dynamic across the time points evaluated in both genotypes. Overall, the expression differences between genotypes were smaller at 4 and 12 DAI than after 30 DAI. Although there is considerable variability in the expression within each TF family, some patterns can be highlighted. Genes from families such as GRFs, HSFs, GRAS, GATA, and bZIP are predominantly overexpressed in the early stages of infection and underexpressed at 30 DAI. However, it seems that in the highly tolerant genotype PI 595099, gene expression reduction occurs earlier than in the susceptible genotype BRS 133. On the other hand, some groups exhibited minor expressions in the early stages of infection. For instance, most members of the BEL1 and KNOX/ELK families of transcription factors were similarly underexpressed at 4 and 12 DAI in both genotypes.

These transcripts were overexpressed much more in PI 595099 than in BRS 133 after 30 DAI. Finally, the largest TF families, such as MYB and bHLH, showed highly variable patterns, mostly overexpressed at 12 DAI and underexpressed at 30 DAI. However, a significant group of bHLH encoding genes was slightly underexpressed at 4 and 12 DAI in both genotypes, with most presenting a dramatic increase in expression in BRS 133 after 30 DAI.

### 2.8. Validation of DEGs with RT–qPCR

A total of 20 differentially expressed genes at both the transcriptome and proteome levels were selected to validate the reliability of the RNA-Seq data through RT–qPCR. The differential expression patterns observed in the RT–qPCR analysis were consistent with those found through the transcriptome analysis (Figure 6). High correlations between the RNA-Seq and RT–qPCR results were observed, with correlation coefficients (R^2^) of 0.92 and 0.97 for the BRS 133 and PI 595099 genotypes, respectively (Appendix A).

### 2.9. GmPR10 Is an Important Soybean Protein against M. incognita

After analyzing and integrating the *in silico* data, one of the genes identified in the present study was selected as a candidate for over-expression in a model plant to increase tolerance to *M. incognita*. The gene choice was based on three criteria (i) differential expression in the transcriptome and proteome; (ii) high levels of expression, detected by RT-qPCR, mainly in the PI 595099 genotype, at some evaluated timepoint (4, 12 or 30 DAI); and (iii) has already been associated with the control of *M. incognita* populations. In this way, among the differentially expressed genes evaluated by RT–qPCR (Figure 6), the high expression level of *GmBetV* gene (also named here *GmPR10*—Glyma.17G030400.1), a plant pathogenesis-related protein capable of increasing plasticity and mechanical barriers in plant responses to pests, such as phytonematodes, was highlighted in the tolerant soybean genotype [68]. The expression level of the *GmPR10* gene observed at 30 DAI in galls of the PI 595099 genotype was approximately 5500 times higher than the non-inoculated control and 4.82 times higher than the observed in the BRS 133 genotype under the same conditions (Figure 6f). Revisiting the *G. max* reference genome assembly (Wm82.a2.v1), 22 different loci were identified as members of this pathogenesis-related protein family (Appendix A). Interestingly, all the homologs of the *GmPR10* gene are present on four different chromosomes (7, 9, 15, and 17), organized *in tandem* repeats with 4–9 copies on each chromosome. The proteins encoded by these soybean genes are small (126–320 amino acid residues) with low molecular weight (13.82–34.81 kDa) and acid pI (4.61–5.17) (Appendix A).

Furthermore, *in silico* analyses demonstrated that among all homologs identified, 11 remained in the same phylogenetic group as the *GmPR10* gene, making them putative paralogs (Appendix A). Structurally, the GmPR10 protein and its 11 paralogs presented three α-helices that flank the seven-stranded anti-parallel β-sheet, highly similar to the structure of the panallergen Ara h 8 from peanuts (*Arachis hypogaea*; PDB ID: 4M9B; Appendix A) [69]. According to the soybean Gene Atlas (https://phytozome-next.jgi.doe.gov/phytomine/begin.do, accessed on 1 March 2022), it is possible to observe high transcript expression levels of these same genes in flowers, leaves, nodules, roots, seeds, and shoots, depending on the evaluated transcript (Appendix A). However, interestingly, all 12 putative paralogs of the *GmPR10* gene (included) showed their highest expression levels in roots when detected (Appendix A).

The effect of *GmPR10* on *M. incognita* reproduction was assessed in three independent transgenic tobacco lines (Figure 7). The number of galls per gram of root in transgenic lines varied from 53.3 to 61.2 in comparison to 126.3 in wild-type (WT) plants. The number of eggs per gram of root was 13,770.7 in WT plants and ranged from 7779.5 to 8006.2 in transgenic plants. These data showed a significant reduction in the number of galls (51.6–57.8%) and eggs (41.9–43.5%) of *M. incognita*. It also explains the decrease (40.4–48.7%) in the reproduction factor in the three-transgenic tobacco lines (81.6–94.7) when compared to WT plants (158.9). Overall, transgenic plants exhibited a reduction of 40.0–58.0% in *M. incognita* reproduction. In parallel, morphological analysis of galls collected at 60 DAI from both WT and transgenic tobacco overexpressing *GmPR10* showed that WT plants presented multiple giant cells filled with dense cytoplasm and contained large nuclei, whereas galls from transgenic lines showed giant cells with few cytoplasm contents, as well as nematodes with altered morphology, and giant cells with thinner cell walls (Figure 8 and Appendix A). All these results corroborated the hypothesis that *GmPR10* overexpression can increase the tolerance of plants to this nematode.

## 3. Discussion

The evaluation of cDNA and protein libraries of two soybean genotypes infected with *M. incognita* provided information of great relevance for understanding several molecular mechanisms of soybean regulated during the interaction with this plant nematode. Even so, considering the classical view of the central dogma of molecular biology, where genetic information flows from the transcription of DNA into RNA, which in turn is translated into a protein, it was expected that the correlation between transcriptome and proteomic data was higher.

This finding is not exclusive to the data mentioned here since the low correlation between transcriptome and proteomic data has already been described in previous studies with other biological systems. For example, transcriptomics and proteomics studies during the interaction of rice and the fungus *Fusarium fujikuroi* showed a correlation ranging from 1.8–2.1% of the total DEGs identified [70]. Several hypotheses can explain the low correlation observed, among which the following stand out: (i) individual characteristics of each transcript that, together or individually, can drastically reduce the translation efficiency, such as week Kazak sequence [71], “codon-bias” [72], as well as the RNA secondary structure [73] and the presence of upstream open read frames (uORFs) in the 5′ non-translated region (5′-UTR) [74]; (ii) differential half-life time between eukaryotic mRNA and proteins [75]; and (iii) difference in the sensitivity of technologies and protocols used for the extraction of total mRNA and proteins, together with evaluation of expression profiles via transcriptome and proteome analysis. Differently from what is observed for mRNAs, the representativeness of the different classes of proteins present in one or more cell types extracted with a specific solvent reduces considerably due to physicochemical and solubility differences [76].

### 3.1. General Molecular Responses to Nematodes in Soybean

A single study has been performed to analyze the transcriptional profiles of soybean (Williams 82, used as a susceptible genotype) infected with *M.*
*incognita* [77]. In a similar approach, our group sequenced the transcriptome of the PI 595099 genotype infected with a phylogenetically closely related species (e.g., *M. javanica*) [47]. In both studies, several genes involved in pathogenesis, cell cycle regulation, and cell wall modifications were differentially expressed. These categories seem to be the key points to understanding the molecular responses in plants against parasitic nematodes. However, those studies evaluated the responses of a single genotype. Therefore, in the present study, a comparison between susceptible and tolerant genotypes was conducted, revealing differences and similarities related to nematode parasitism. Primarily, it is important to clarify that explaining a determined molecular response strictly by the nematode influence would be inaccurate. Most expression changes in our study are presumably associated with plant development over time. Nevertheless, some processes, protein families, and metabolic pathways have been described in previous works to participate in the defense against nematode infection [78,79]. Genes belonging to these groups can be divided into two categories for the purposes of this study. First, a set of genes showing the same patterns in the susceptible and tolerant genotypes, which are basal defense mechanisms and conserved resistance traits. Second, genes with opposite expression patterns between genotypes, unique features of a given genotype, or even differences in the intensity of their expression. This last group is highly relevant to the objective of this research since such genes could help to elucidate the molecular mechanisms behind the tolerance to *M. incognita* exhibited by PI 595099.

The first difference between the genotypes evaluated was that PI 595099 presented more DEGs and DEPs than BRS 133 (Appendix A). Furthermore, transcripts and proteins differentially expressed only in PI 595099 were approximately half of the total in most cases, while unique DEGs and DEPs for the susceptible genotype were less than 20% in almost all treatments. Hence, infection with *M. incognita* has a larger effect on the molecular regulation of genes and proteins in the tolerant genotype. This trend has also been reported for other transcriptomes of soybean genotypes infected with another type of nematode, the soybean cyst nematode (*Heterodera glycines*) [80,81,82]. At the protein level, however, the number of DEPs seems to be similar to other soybean studies [83,84,85].

From a functional view, most responses were observable only at 30 DAI (Appendix A). Thus, plant development greatly affects how many responses are triggered by RKN infection. Interestingly, several genes involved in nuclear division and replication are strongly underexpressed in PI 595099. Since RKNs induce abnormal cell proliferation in the host to serve as feeding sites, an efficient defense mechanism would be reducing the rate at which cells divide themselves [9]. Additionally, PI 595099 overexpressed genes that inactivate cytokinins (CKs), proteins that favor cell division and growth [86]. In addition, the activity of GTPases at the transcriptional and protein levels is reduced in the tolerant genotype. Some GTPases are implicated in signaling pathways required to maintain the balance between cell differentiation and cell division [87]. Therefore, this evidence indicates the presence of a mechanism in PI 595099 (lacking BRS 133) to suppress cell proliferation at the transcriptional level. Moreover, homologs to the PI3K/AKT/mTOR pathway, a signaling pathway with a crucial role in cell cycle regulation in animals, presented several genes that were over- and underexpressed in PI 595099 (Figure 2). However, whether these genes have similar functionality in soybean remains to be investigated [88].

Notably, previous studies have focused on describing early responses of most tolerant genotypes to parasitic nematodes; thus, defense mechanisms that produce an extensive reaction at the beginning of the parasitism process have been identified. Examples of such processes are the induction of receptor-like kinases (RLKs) and resistance genes (*R* genes), such as *Rhg1* (resistance to *H. glycines* 1) [82], or the activation of calcium/calmodulin-mediated signaling and LRR (leucine-rich repeats) proteins [80]. In contrast, in the present study, later responses of the tolerant genotype were mainly related to cushioning the effects of the pathogen and restricting its reproduction rather than directly inducing *R* genes.

### 3.2. Transcription Factors Orchestrating Soybean Responses to M. incognita Infestation

As previously mentioned, the TFs are key regulatory proteins that modulate the expression of genes involved in several plant processes, such as defense signaling pathways against biotic and abiotic stresses [89,90]. The major TF families MYBs, WRKY, AP2, and bHLH are known to play important roles in plant responses to nematode parasitism. Previous works validating the biological role of some TFs have shown that they can act as positive or negative regulators in the plant defense regulatory network against nematodes. For instance, the knocking down of *SlWRKY72a* and *SlWRKY72b* genes led to decreased *Mi-*1-mediated resistance and basal defense against *M. incognita* in *S. lycopersicum* and *A. thaliana* [91]. Likewise, *SlWRKY3* loss-of-function tomato mutants showed higher infection of *M. javanica*, while *SlWRKY3* overexpression decreased the infection [92]. In contrast, the knocking down of *WRKY23* gene in *A. thaliana* greatly reduced its susceptibility to the nematode *H. schachtii* [93]. Furthermore, transgenic *A. thaliana* plants overexpressing RAP2.6, an AP2/ERF transcription factor, exhibited enhanced deposition of callose in syncytia and reduced susceptibility to *H. schachtii* [94]. Moreover, altered roots and shoots’ morphology, as well as enhanced susceptibility to *H. schachtii* were observed in *A. thaliana* lines overexpressing bHLH25 and bHLH27 TFs [95]. Based on these findings and the results reported here, we speculate that some of the TFs differentially expressed in the infected soybean genotypes might be involved in plant-nematode interactions. However, members of the major families of TFs in our transcriptome presented highly variable patterns of expression. This indicates that regulation of defense responses does not occur at the family level but could be related to the expression of specific alleles and protein variants within a TF family. Thus, the DEGs identified in our study, with larger discrepancies between genotypes, should be considered for future functional studies as promising genes related to traits of soybean that confer resistance to RKN.

On the other hand, some minor TF families showed consistent differences in expression levels of all members of the given family. For instance, the expression of GRFs exhibits a faster reduction in the tolerant genotype than in the susceptible genotype. It has been demonstrated that microRNA-mediated knockdown of several GRF genes regulates syncytium development in a model of cyst nematode infection [96,97]. This rapid downregulation of GRFs in PI 595099 suggests a similar resistance mechanism as already described for cyst nematodes. Sequencing and validation of miRNAs expressed in the soybean-*M. incognita* interaction could aid in elucidating these mechanisms. In conclusion, future studies should be carried out to determine whether the TFs found in our study participate in the plant defense response and elucidate the molecular mechanisms by which they modulate soybean transcriptional changes during *M. incognita* parasitism.

### 3.3. The Phenylpropanoid Pathway and GmPR10 Activity Are Important Soybean Mechanisms of Defense against M. incognita

Nematode infections induce extensive gene expression reprogramming in host roots. Most of the DEGs were strongly related to defense pathways, whose responses led to the accumulation of secondary metabolites. Under this view, the phenylpropanoid biosynthetic pathway is one of those with a higher number of target genes differentially regulated during nematode infection [98,99,100,101].

The metabolism of phenylpropanoids was particularly enriched in both the proteome and transcriptome. The susceptible genotype exhibited underexpression of genes belonging to this pathway at 4 and 30 DAI at the transcriptional and protein levels. On the other hand, in PI 595099, DEGs underexpression was observed at 30 DAI, while proteins were overexpressed at the same time and 12 DAI. These genes were equally represented in susceptible and tolerant genotypes, which suggests a central role in the early soybean response against *M. incognita* [98,102].

The phenolic compounds in plants are divided into two main classes, sharing the initial steps of biosynthesis. The amino acid phenylalanine is converted into cinnamic acid (Figure 4a, step 1) and, subsequently, into p-coumaric acid (Figure 4a, step 2), which is finally converted into 4-coumaroyl-CoA (Figure 4a, step 3). These reactions are catalyzed by three important enzymes involved in phenolic compound metabolism: phenylalanine ammonium lyase (PAL), C4H, and 4CL. In the tolerant genotype, the enzymes C4H and 4CL were overexpressed during nematode infection, suggesting a more efficient activation of pathogen-responsive pathways, which could be associated with tolerance mechanisms. From the core of phenylpropanoid-acetate metabolism, several reactive secondary metabolites are produced, such as chalcones, flavones, flavonols, flavonoids, anthocyanins, tannins, aurones, isoflavonoids, and pterocarpans [103,104,105,106]. These are associated with basal and defensive roles, such as flower pigmentation, control of auxin transport, ROS-scavenging, chemoattractants, and plant–pathogen signaling- and defense-associated molecules [107].

Our analysis of GO enrichment revealed four main pathways of secondary active metabolites derived from phenylalanine: (i) biosynthesis of glyceollin, a pterocarpan-derived phytoalexin (Figure 4b); (ii) synthesis of naringenin (Figure 4c), a flavanone; (iii) anthocyanin-derived biosynthesis (Figure 4d); and (iv) the biosynthesis and polymerization of lignin (Figure 4e). The plant nematode parasitism process inflicts mechanical damage on roots [107], triggering the production of defensive compounds responsive to a broad range of molecular signals associated with pathogenesis, such as the hormones jasmonic acid, salicylic acid, ethylene, and auxin, as well as ROS accumulation [79,108]. Several studies have reported that more tolerant genotypes display a markedly higher expression of genes involved in producing these active metabolites, including phenylpropanoid-derived flavonoids [107]. Phenylpropanoids take part in wound- and defense-associated responses in plants [109], constituting a valuable target for the development of more tolerant genotypes against phytonematodes. Our investigation reported that C4H and 4CL, crucial enzymes in 4-coumaroil-CoA biosynthesis, were overexpressed in the tolerant genotype, as expected (Figure 4a). 4CL is reported to be overexpressed in response to cyst-nematode infection in soybean (Williams82) [110] and displays higher expression levels in the roots of more tolerant soybean plants infected with *H. glycines* and *M*. *incognita* [111,112].

4-Coumaroil-CoA acts as a hub in several branches of phenylpropanoid-derived metabolites. For example, the synthesis of phytoalexins, anthocyanins, and lignin, which are associated with nematode tolerance, depends on 4-Coumaroil-CoA as a primary substrate. Interestingly, at least one gene of each pathway was overexpressed in the tolerant genotype, reinforcing the suggestion of an active phenylpropanoid pathway as a significant tolerance-associated mechanism highlighted by our transcriptome and proteomic analysis.

The tolerant genotype also showed higher CHS and isoflavone reductase (IFR) expression than the susceptible genotype. These enzymes belong to the flavanone and phytoalexin branches and catalyze limiting steps in naringenin and glyceollin production. Defense compounds from isoflavonoid and pterocarpan classes are commonly associated with nematicidal activity in several crops, such as soybean [113,114,115], cowpea and common bean [116,117], and rice [118]. In soybean, it has been reported that *M*. *incognita*-inoculated roots exhibit a hypersensitive response and accumulate different types of glyceollin, a product of the isoflavonoid branch [113,114,115]. Additionally, they can act in broad stages of nematode development, impairing egg hatching [98,119], nematode movement, migration, and feeding site establishment [107,109]. Glyceollin can also inhibit oxidative processes from respiration in *M*. *incognita* and immediately accumulates around the head region of cyst nematodes (*H*. *glycines*) in more tolerant soybean genotypes, which is not observed in the tissues of susceptible soybean roots [113,114]. Both enzymes were reported to be overexpressed in more tolerant alfalfa and soybean plants infected with different RKN species, as well as in response to cyst-nematodes in the Williams 82 susceptible genotype [110,111,112,120].

The CHS enzyme also drives the biosynthesis of naringenin, a flavanone with egg anti-hatching activity [119], and the primary substrate of F3′H for anthocyanin biosynthesis (Figure 4c,d). In addition to F’3H, the enzyme LDOX, which catalyzes the final step of anthocyanin production, is also overexpressed in the tolerant genotype. Anthocyanins are an important class of flavonoid-derived pigments that mainly exhibit UV- and ROS-protective characteristics in addition to antimicrobial activity in plants [103]. Plants produce ROS not only to oppose the parasites in extremely oxidative environments but also to activate defense responses and programmed cell death (PCD) under adverse conditions [94]. In addition to secondary metabolites, ROS production is dependent on hormones, especially salicylic acid, which is the main stimulator of the hypersensitive response [121,122]. Despite the crucial role of ROS production in nematode-defensive pathways, they also impair plant cell survival. Thus, the accentuated production of ROS by the tolerant genotype in contrast with the production of ROS-protective compounds might be a part of a homeostasis mechanism in these plants.

Two enzymes of lignin precursor biosynthesis and polymerization were found to be overexpressed in the tolerant genotype. CCoAMT catalyzes the early reactions of coniferyl- and sinapyl-derived monomers that are polymerized in G and S lignin, respectively. Lignin is an amorphous heteropolymer resulting from the oxidative coupling of alcohols catalyzed by reactive cycles of peroxidases (POX) and LCCs. Lignin deposition in cell walls increases their hydrophobicity, as well as their tolerance to mechanical stretching [123], providing a physical barrier against nematode attack [124].

In addition, the phenylpropanoid-mediated soybean response can be directly associated with the activity of proteins from the PR10 family. Despite the protective role assigned to members of this protein family, especially in response to biotic and abiotic stresses, their mechanism of action is still not fully understood [125]. Different studies report that members of the PR10 family can present RNase activity [126,127,128,129,130,131,132], can interact with phytosteroids [133,134,135], fatty acids, cytokinins, and brassinosteroid analogs; regulate or directly contribute to the biosynthesis of sporopollenin, phenolics, flavonoids, and alkaloids [134,136]; can be modified by the addition of glutathione [129,137] or phosphate groups [138], or even convert emodin to hypericin in vitro [139]. In addition, studies that characterized the promoter region of *AoPR10* gene from *Asparagus officinalis* demonstrated that the *pAoPR10*::*uidA* transgene in *A. thaliana* was responsive to oxidative signals/stresses, such as phenylpropanoid accumulation, wounding, pathogen infection and treatment with hydrogen peroxide (H_2_O_2_) [140].

Members of the PR10 family can also have a negative effect on nematodes by inhibiting and/or degrading enzymes present in the digestive tracts and cuticles of these pathogens [136]. For example, CpPRI protein purified from *Crotalaria pallida* roots demonstrated nematostatic and nematicidal effects in bioassays with *M. incognita*. Just six hours after administration, the CpPRI protein had already spread throughout the nematode’s body (J2 stage) and, depending on the concentration administered, presented a nematicidal effect forty-eight hours after administration, reaching up to 95% mortality. The high lethality mediated by CpPRI may be related to its papain inhibitory activity and inhibition of intestinal cysteine proteinases present in the J2 intestinal tract [141]. Further transcriptomic studies with *Pinus thunbergii* during interaction with the pine wood nematode (PWN) *Bursaphelenchus xylophilus* showed the synchronized overexpression of *PtPR10* and peroxidase genes in pine trees resistant to PWN, which indicates the induction of *PtPR10* overexpression by ROS produced during plant-nematode interactions. In this way, *PtPR10* can show protease activity against some proteins secreted by PWN, such as cellulases, β-1,3-glucanase, and pectate lyases [142].

Thus, the differential expression of the *GmPR10* gene observed both in the omic data and in the RT–qPCR experiments (Figure 6f), associated with the protective effect against *M. incognita* observed in transgenic tobacco overexpressing the *GmPR10* gene (Figure 7 and Figure 8), strongly indicates that this pathogen-related protein plays an important role in the high tolerance of the PI 595099 genotype to *M. incognita*. Furthermore, it is possible to infer that the regulation of *GmPR10* expression can be mediated by ROS or other compounds produced in the phenylpropanoid pathway, similar to what was observed by Mur and collaborators during the characterization of *AoPR10* gene promoter in *A. thaliana* [140]. This hypothesis is supported by the identification of a smaller number of differentially expressed genes associated with the phenylpropanoid pathway in the susceptible genotype (BRS 133) than in the highly tolerant genotype (PI 595099). On the other hand, the hypothesis that GmPR10 protein can regulate phenylpropanoid biosynthesis can also be considered plausible, but in this case, the lower number of differentially expressed genes associated with the phenylpropanoid pathway in the susceptible genotype would be a consequence of the low expression of the *GmPR10* gene. In both hypotheses, the activity of GmPR10 protein and phenylpropanoids biosynthesis would act together in soybean for an effective response against *M. incognita*. Therefore, the synchronized overexpression of soybean *PR10* genes and genes associated with phenylpropanoid biosynthesis can improve the protective effect against infestation by *M. incognita* more so than if these elements were overexpressed independently. To this end, synthetic biology technologies such as CRISPR/dCas9 with multiple single-guide RNAs (sgRNAs) can be considered important biotechnological tools since it is possible to modulate the expression of different genes by recruiting transcriptional activators/repressors to their respective gene promoters [143].

In this way, understanding the mechanism of nematode control associated with more tolerant accessions/genotypes, such as the soybean PI 595099 genotype, allows the development of environmentally friendly pest management programs and modern strategies for molecular breeding. Collectively, our results highlight the pivotal role of oxidative metabolism and phenylpropanoid biosynthesis as the molecular basis of parasitism defense and identify suitable targets for biotechnology intervention in superior crop production.

## 4. Material and Methods

### 4.1. Bioassay Conditions

The genotypes selected for this study comprised BRS 133, as susceptible to *M.*
*incognita*, and PI 595099, as highly tolerant to the same pest [144]. The selection of the soybean genotypes was based mainly on the RF values. First, three independent experiments, each with 10 plants of each genotype, were performed. Soybean seeds from each genotype were germinated on filter paper (Germitest) for seven days in a growth chamber at 25.0 ± 1.0 °C and 100.0% relative humidity (RH). Then, seedlings of the same size were planted in conical tubes with 300 mL of soil (1:1 sand:clay). After reaching the V2 stage of development, each soybean plant was inoculated with 1350 *M. incognita* nematodes in the J2 stage of development. The *M. incognita* race 1 population was propagated in tomato plants (*Solanum lycopersicum* L. cv. Santa Clara) for three months under greenhouse conditions. Tomato-infected roots were blended in 0.5% NaOCl, and eggs were extracted according to the Hussey and Barker (1973) method [145]. Eggs of *M. incognita* were retained in the 500-mesh sieve and counted in a Peters chamber.

At 60 DAI (representing two nematode life cycles) with *M. incognita* race 1, the eggs were extracted from the inoculated roots, also according to the Hussey and Barker (1973) method [145], then later counted in a Peters chamber and for determination of the RF of both soybean genotypes (RF = Fp/Ip, where *Fp* = number of eggs and *Ip* = initial number of inoculated J2).

A second bioassay was conducted using the same methodology as that employed for RF determination for the collection of biological material for use in the transcriptome, proteome, and RT–qPCR experiments. This second bioassay was also performed with the BRS 133 and PI 595099 genotypes, where samples of inoculated roots at 4, 12, and 30 DAI were collected, as well as from non-inoculated plants (time zero, stage V2 of development). Four biological samples were collected in each condition, and each biological sample consisted of samples from 10 plants. Regarding the collection of biological material, at 4 DAI, only the root tips that showed protuberance with a strong indication of the nematode entry site were sectioned and collected; at 12 and 30 DAI, only galls were selected. Once collected, each biological sample was macerated in liquid nitrogen until a fine powder was formed. The samples were kept at −80 °C until they were subjected to protein and total RNA extraction procedures.

For the bioassays with transgenic tobacco plants overexpressing the *GmPR10* gene, T_2_ plants were inoculated with approximately 1000 hatched second-stage juveniles (J2) of *M. incognita* race 1. At 60 DAI, tobacco roots were collected, and the number of galls, eggs, and *M. incognita* RF was determined in the same manner described above for soybean. The bioassays with tobacco were performed with one non-transformed WT group and three transgenic lines overexpressing the *GmPR10* gene.

All results were analyzed using Tukey’s post-hoc test with multiple comparisons and one-way ANOVA. All statistical analyses were carried out using GraphPad Prism 7.0 software.

### 4.2. Morphological Analysis

For the two contrasting soybean genotypes (BRS 133 and PI 595099), the samples inoculated with nematodes were evaluated to describe the morpho-physiology of the cell-nematode interaction (soybean—*M. incognita*). The first parameter evaluated was the PEf. For this, 10 roots of both inoculated genotypes at 4 DAI with *M. incognita* were stained with acid fuchsin, which allowed the visualization of the nematodes that were successful in penetrating the roots. In this procedure, the roots were previously incubated in 5.25% NaClO for whitening. After washing in distilled water, the roots were boiled in 0.25% acid fuchsin. After another round of washing with distilled water, the roots were boiled again in 100% acidified glycerol, where they were stored until the nematodes that penetrated the roots were counted. The internalized nematodes were counted using a stereoscopic magnifying glass. The calculation of PEf was made from the following formula: PEf = (Ni/1350) × 100, where *Ni* = number of internalized nematodes, and 1350 is the number of previously inoculated J2 nematodes.

For the initial morphological analysis of two contrasting soybean genotypes (BRS 133 and PI 595099) inoculated with *M. incognita* (race 1), root samples inoculated with the phytonematode were collected at 4, 12, and 30 DAI and fixed in glutaraldehyde for 8 days at 4 °C (under slight agitation). For histological analysis with transgenic tobacco overexpressing the *GmPR10* gene inoculated with *M. incognita* (race 1), root samples (galls) were collected at 60 DAI and treated in the same manner as described below. Later, all the samples were dehydrated in serial dilutions of cold ethanol (15%, 30%, 50%, 75%, 85%, and 100%). Then, the fixed and dehydrated samples were embedded in Technovit^®^ 7100 plastic resin according to the manufacturer’s instructions (EMS^©^, Hatfield, UK, Catalog No. 14,653). After the polymerization of the resin, each block was sectioned in a microtome with semithin sections of 5 µm and floated in drops of water. Each slide containing an average of 40 histological sections was incubated at 42 °C for 16 h so that the sections could adhere well to the slides. Finally, each slide was stained with 1% toluidine blue and sealed with Depex (Sigma^©^, Oliver Township, MI, USA) for further analysis.

### 4.3. Total RNA Extraction and Transcriptome Analysis

Total RNA extractions from three biological samples from each condition (control, 4, 12, and 30 DAI) were made according to the manufacturer’s instructions of the ReliaPrep^™^ miRNA Cell and Tissue Miniprep System kit (Promega, WI, USA, CAT No Z6212). The TruSeq^™^ SBS v5 protocol was used for library construction (Illumina, San Diego, CA, USA) and subsequent RT–qPCR analysis. The 24 resulting libraries were sequenced on an Illumina HiSeq 4.000 at the University of Illinois (USA). cDNA library construction was performed to produce PE reads with 150 nucleotides each. Adaptors and low-quality raw sequences (Phred < 28) were trimmed using Trimmomatic v0.39, based on the Phred 33 score assessed by FastQC, using the following parameters: CROP: 140, HEADCROP: 10, SLIDINGWINDOW: 4:15, and MINLEN: 100. All processed reads were quantified by pseudo-alignment to the predicted transcripts of the *G. max* assembly genome (Wm82.a2.v1) (https://phytozome.jgi.doe.gov/, accessed on 1 September 2020) using Kallisto v0.46.1 [146], with the default settings to obtain transcript counts and abundances. Lastly, statistical detection of differential expression was determined by using DESeq2 v1.26.0 [147]. Soybean transcripts were considered as DEGs when their relative expression levels showed an FC value between inoculated and control samples [log_2_(FC) > 2 or <−2] and adjusted *p-value* (*FDR* < 0.05).

To assess the expression levels of the *GmPR10* and *eGFP* genes in transgenic tobacco, total RNA was isolated from ten independent transgenic roots 60 DAI with *M. incognita* race 1 using TRIzol Reagent (Invitrogen, Waltham, MA, USA) according to the manufacturer’s instructions. The RNA concentration was estimated using a spectrophotometer (NanoDrop 2000, Thermo Scientific, Waltham, MA, USA), and RNA integrity was evaluated by 1% agarose gel electrophoresis.

For cDNA synthesis, all RNA samples evaluated in this study were treated with RNase-free DNase I (Promega, Madison, WI, USA) according to the manufacturer’s instructions. Then, 1 μg of DNase-treated RNA was used for reverse transcription and (RT) cDNA synthesis using oligo-NVdT_30_ primers and SuperScript III RT (Life Technologies, Carlsbad, CA, USA) according to the manufacturer’s instructions. All synthesized cDNA was diluted (1:20) for further RT–qPCR experiments.

### 4.4. Quantitative Reverse Transcription PCR (RT–qPCR)

Quantitative reverse transcription PCR was performed to validate the gene expression of 20 overexpressed genes in at least one condition of both the transcriptome and proteome analysis. This technique was also applied to evaluate the expression levels of the *GmPR10* and *eGFP* genes in transgenic tobacco roots. Primers were designed by Primer 3 Plus [148] software and checked for the presence of putative amplicons from 120 to 200 bp and a melting temperature (T_M_) of 60.0 ± 2.0 °C (Appendix A). The primers used to measure the expression of the *GmPR10* and *eGFP* genes in transgenic tobacco roots were *GmBetV* and *eGFP,* also listed in Appendix A. To establish the normalization factor, two reference genes were used for soybean root samples (*GmERF1A* and *GmCYP2*) and two for transgenic tobacco (*NtACT1* and *NtL25*) (Appendix A). All experiments were carried out in experimental and biological triplicates. The quantitative real-time PCR amplifications were performed using the ABI Real-Time PCR System 7500 Fast (Applied Biosystem^©^, Waltham, MA, USA) thermal cycler with a comparative cycle threshold (2^ΔΔCt^). Rox Plus SYBR Green Master Mix 2X (LGC^©^, Cotia, SP, Brazil) was combined with 4.0 μM of each primer (sense and antisense) and 2.0 μL of cDNA (20-fold dilution) for each experimental condition. The PCR cycling conditions were 95 °C for 15 min to activate the hot-start Taq DNA polymerase, 40 cycles at 95 °C for 30 s, 60 °C for 30 s, and 72 °C for 3 min (final extension). The raw fluorescence data for all runs were imported into the Real-Time PCR Miner software [149] to determine the C_t_ value and the PCR efficiency. The C_t_ values were converted by qBASE v1.3.5 software [150]. The statistical analysis was performed using the REST 2009 software (Relative Expression Software Tool—Qiagen^©^, Hilden, Germany) [151].

### 4.5. Protein Extraction and Proteome Analysis

For proteomics experiments, the total fraction of soluble proteins was extracted using the phenol-chloroform extraction protocol [152]. Proteins were extracted from each biological sample collected in the “Bioassays Conditions” step in the methods section. The composition of the extraction buffer was 0.7 M sucrose, 0.5 M Trizma base, 63.5 mM EDTA, 10.0 mM KCl and 40.0 mM dithiothreitol (DTT). For each 100 mg of previously macerated root tissue, 750 µL of extraction buffer (1 volume) was added, followed by incubation at room temperature for 15 min. After the incubation period, 1 volume of phenol-chloroform was added, and the mixture was kept under vigorous stirring for 15 min. Then, each sample was centrifuged for 3 min at 13,000× g at room temperature. After phase separation, the phenolic phase was collected and precipitated with 5 volumes of 0.1 M ammonium acetate in methanol at −20 °C overnight. The next day, each sample was centrifuged for 3 min at 13,000× g at 4 °C, and the pellet was dried at room temperature for 5–10 min.

The dried protein samples were resuspended in denaturation buffer (6 M urea, 2 M thiourea, 1X cOmplete^®^ Mini Protease Inhibitor Cocktail). The samples were reduced by 10 mM DTT at room temperature for 30 min and alkylated by using 20 mM iodoacetamide in the dark for 30 min. Qubit quantification was carried out to determine the protein concentration. One microliter of LysC enzyme was added to each sample and incubated for approximately 3 h at 28 °C, after which the samples were diluted 10 times with 20 mM TEAB buffer. After dilution, trypsin enzyme was added to the samples in a 1:50 ratio (trypsin to protein ratio) and incubated at 28 °C overnight.

The tryptic peptides were desalted by using an HLB column (Waters Oasis). The purified peptides were quantified by using an amino acid analyzer (AAA), as described previously. One microgram of each sample was loaded onto a Proxeon Easy-nLC system (Thermo Fischer Scientific, Odense, Denmark) coupled with a Q Exactive™ HF Hybrid Quadrupole-Orbitrap™ Mass Spectrometer (Thermo Fischer Scientific, Odense, Denmark) for protein identification with 75 min of gradient time using an 18 cm long column. The top twenty most abundant precursor ions selected by the quadrupole during the initial MS scan were subjected to fragmentation using high-energy collision dissociation.

Each of the samples was run in three technical replicates, and the raw files generated were processed with MaxQuant software (version 1.6.0.16) and its built-in search engine, Andromeda, for peptide identification with default parameters [153,154] using *G. max* version Wm82.a2.v1 (https://phytozome.jgi.doe.gov/, accessed on 1 September 2020). The label-free quantification (LFQ) intensities were then imported into the Perseus program from protein groups to log transform the data [155]. The selection criteria were protein identification in at least two biological replicates. The list of selected proteins was exported from Perseus, and statistical analysis was performed using PolySTest, where in one comparison, all the groups were analyzed against the control groups (day zero, not inoculated roots) for both susceptible and tolerant soybean genotypes. The DEPs were selected following a *q*-value cutoff of 0.05 for statistical significance of regulation, whereas the expression cutoff was log_2_ (FC) < −1 or >1.

### 4.6. Functional Enrichment of Gene Ontology and Metabolic Pathways

To provide a curated source of features for our predicted proteins, the annotation DEG and DEP datasets were based on the Wm82.a2.v1 soybean assembly and PFAM database (http://pfam.xfam.org/, accessed on 5 September 2020). The GO and Kyoto Encyclopedia of Genes and Genomes (KEGG) pathway enrichment analyses were performed for each set of DEGs and DEPs independently, considering a set as a combination of a given genotype in a specific treatment time with a determined sense of regulation (overexpression or underexpression). The whole genome was used as background in all cases. Both functional enrichments were calculated using a hypergeometric test (HGT). The logical values were the GO and KEGG IDs in the respective analyses. For gene ontology, HGT was performed using FUNC software [156], while for metabolic pathways, a standalone version of HGT was executed in R software v4.0.3 using the R Stats Package (https://www.r-project.org/, accessed on 9 September 2020). The cutoff for statistical significance was an *FDR*-corrected *p* value < 0.05.

### 4.7. GmPR10 In Silico Analysis

Sequences and features of soybean genes were retrieved from the *JGI G. max* Wm82.a2.v1 dataset [157,158] from the Phytozome v.13 database [159]. Initially, 21 protein sequences homologous to GmPR10 (*Glyma.17G030400.1*) were identified with the BLASTp *in silico* tool [160]. The molecular weight (MW) and isoelectric point (pI) of the GmPR10 protein and its paralogs were determined on the *Compute pI/Mw* online platform. (https://web.expasy.org/compute_pi/, accessed on 3 March 2022) [161]. The 11 paralogs of GmPR10 protein were selected with OrthoMCL v6.11 using default parameters [162]. Conserved domains in the protein sequences were identified using the HMMER prediction server [163], and annotations were confirmed by the NCBI CDD database [164]. The organ- and tissue-specific expression of the *GmPR10* gene and its 11 different paralogs was represented by a heatmap generated with the PhytoMine tool (https://phytozome.jgi.doe.gov/phytomine/begin.do/, accessed on 1 March 2022) using gene expression data from the Gene Atlas version 2 experimental groups. For phylogenetic analysis, the protein sequence of the GmPR10 protein was aligned with that of its paralogs and the selected outgroup (major allergen Pru av 1-like from *Eucalyptus grandis*; XP_010064159.2) with MAFFT software v7.402, *-auto* option [165]. Given the aligned coverage and the high level of identity/similarity, the resulting alignment was directly used as input for analysis via the *maximum likelihood* method with the software Randomized Axelerated Maximum Likelihood (RAxML; v8.2.12) with options *-#autoMRE* (number of bootstraps required determined by software) and *-m PROTGAMMAAUTO* [166]. The phylogenetic tree was analyzed and annotated using the online tool Interactive Tree of Life (iTOL; v6.5.2; https://itol.embl.de/, accessed on 11 March 2022) [167]. To confirm the relationship between the GmPR10 protein and its paralogs, the structure of each selected protein sequence was determined by homology with structures already deposited in public databases. The sequences were individually modeled using the model deposited in the Protein Database (PDB 4M9B, crystal structure of Apo Ara h 8), which showed greater sequence coverage (greater than 0.93) and greater identity (greater than 50.32%). High-quality models were generated with pairwise target-template alignment submitted as input to the SWISSMODEL server (https://swissmodel.expasy.org/, accessed on 13 March 2022) [168]. Structural models were superimposed with the SALIGN online web server (https://modbase.compbio.ucsf.edu/salign/, accessed on 15 March 2022) [169].

### 4.8. GmPR10 Overexpression in Transgenic Tobacco

To overexpress the *GmPR10* gene in transgenic *Nicotiana tabacum*, the binary vector pPZP-GmPR10-eGFP-HygR was synthesized and assembled by Epoch Life Science (Missouri City, TX, USA). The construction had a 477 bp *GmPR10* coding sequence (Glyma.17G030400.1) driven by the cauliflower mosaic virus *CaMV35S* promoter. The *hptII* gene (hygromycin resistance) was used as a selection marker under the control of the *AtUbi3* promoter. The expression of the enhanced green fluorescent protein *(eGFP*) gene was also driven by the *CAMV35S* promoter and used as a molecular marker protein for the detection of transgenic plants. The resulting construction was checked by sequencing.

*Agrobacterium tumefaciens* strain GV3101 was transformed with the pPZP-GmPR10-eGFP-HygR gene construct. Transformed bacteria were grown in Luria Broth (LB) media supplemented with rifampicin (100 µg.mL^−1^), gentamicin (50 µg.mL^−1^) and kanamycin (100 µg.mL^−1^) for 16 h at 28 °C (OD_600_ of 0.8) [170]. *N. tabacum* L. (var. Petit Havana) was transformed and multiplied in vitro employing the leaf disc method [171]. Sterile *N. tabacum* leaf explants were incubated in *A. tumefaciens* culture for 1 h in the dark at 28 °C. Leaf fragments were dried and transferred to plates containing solid MS medium (pH 5.7) supplemented with 1% sucrose, 1X Gamborg’s vitamin solution, and 1.0 mg L^−1^ of 6-benzylaminopurine (BAP) and maintained for 2 days in the dark at 28 °C. Leaf fragments were transferred to solid MS plates (pH 5.7) containing 1% sucrose, 1× Gamborg’s vitamin solution, 1.0 mg L^−1^ of BAP, 0.1 mg L^−1^ of the hormone naphthalene acetic acid (NAA), 100 mg L^−1^ of cefotaxime and 50 mg L^−1^ of hygromycin for callus induction [170,171].

Seedlings were transferred to Magenta GA7 boxes containing solid MS medium: MS—4.33 mg L^−1^, sucrose—10 g L^−1^, pH 5.7, phytagel—3.22 g L^−1^, and hygromycin—50 mg L^−1^ and grown for 4 weeks. The regenerated shoots were excised at the base and transferred to Magenta GA7 boxes containing MS rooting medium (MS—4.33 g L^−1^, sucrose—10 g L^−1^, pH 5.7, phytagel—3.22 g L^−1^, and indol-butyric acid—1 mg L^−1^). Non-transformed explants were cultivated onto the same media with or without hygromycin as negative and positive controls, respectively. Hygromycin-resistant plants were acclimatized in pots containing commercial substrate and grown in a greenhouse at 25 ± 10 °C and 50% humidity. Ten plants in the T_1_ and T_2_ generations were initially selected by in vitro screening of tobacco seeds germinated in MS medium (MS—4.33 g mg L^−1^, sucrose—10 g L^−1^, pH 5.7, phytagel—3.22 g L^−1^) supplemented with hygromycin (50 mg L^−1^) [170,171].

For molecular characterization, genomic DNA was extracted from leaf tissue using the CTAB method [172]. Transgene was detected by PCR using the set of primers GmBetV (193 bp) (Appendix A). PCR was carried out in a 20 µL reaction mixture containing 0.25 mM DNTP solution, 1.5 mM MgCl_2_ solution, 0.2 pmol of each primer, 1 U of Taq DNA polymerase (Invitrogen, Carlsbad, CA, USA) and 100 ng of DNA. The cycling conditions were as follows: an initial denaturation step (95 °C, 5 min) followed by 40 cycles of annealing (95 °C, 20 s), extension (60 °C, 20 s) and denaturation (72 °C, 20 s), and a final extension step (72 °C, 5 min).

## 5. Conclusions

We highlighted in the previous paragraphs some of the most significant responses of soybean genotypes to nematode infection. However, the expression profiles observed in other hosts infected with *M. incognita* are valuable information to understand whether conserved tolerance/resistance mechanisms or host-specific traits are crucial factors in generating nematode-resistant soybean cultivars. As an example, major GO categories enriched in our study, such as oxidation-reduction process, response to stimulus, biosynthesis of secondary metabolites, carbohydrate metabolism, and stress-related genes, have been observed in previous transcriptomes of cucumber, tobacco, alfalfa, and cotton infected with *M. incognita* [51,62,63,66], suggesting that these processes might be characteristic of constitutive plant responses to *M. incognita* infection. In contrast, our results indicated that genes related to the plant hormone signaling pathway, which have been associated with the plant response to nematode infection [48,60,65], were not significantly overrepresented during the soybean-*M. incognita* interaction.

Recent studies in rice, sweet potato, and cucumber have suggested that cultivars resistant to *M.*
*incognita* exhibit expression profiles consisting of large sets of genes involved in complex signaling pathways. Among these, the Ca^2+^ pathway, phytohormones, and protein kinases are the core of responses in resistant cultivars. A decreased gall formation and size rate were typically related to these responses [173,174,175,176]. Interestingly, in our work, tolerant soybean seems to trigger responses independent of the action of phytohormones. The DEGs and DEPs found in our study indicate mechanisms of resistance related to the production of secondary metabolites that limit the reproduction of the parasites. This phenomenon has been proven in rice and patchouli (*Pogostemon cablin*), hosts in which systemic defense against *M. incognita* is not correlated with hormone accumulation but is highly dependent on stimulation of the phenylpropanoid pathway [102,177].

This study also comprehensively described the gene expression and protein production profiles present during the progression of soybean-*M. incognita* interactions. We identified the pivotal genes and proteins that mediate constitutive defenses against nematodes in susceptible and highly tolerant genotypes, as well as the crucial mechanisms that can explain resistance traits of the tolerant genotype. The most remarkable of these was the differential regulation of the phenylpropanoids pathway and identification of the core enzymes involved in plant defense against biotic stress. In addition, we highlighted the similarities and differences in the regulated pathways and protein families found in the transcriptome and proteome of soybean in response to root-knot nematode *M. incognita* infection. In summary, it is clear that during the soybean-nematode interaction, an oxidative “burst” is triggered by the plant counterpart. Even though the hypersensitivity response (HR), which culminates in cell death of the parasitized cells, has not been observed in soybean galls, it can be concluded that compounds such as H_2_O_2_, ROS, and nitric oxide (NO) [178] orchestrate direct damage responses to the nematode and indirect responses, such as improvement of phenylpropanoid biosynthesis, as well as the overexpression of pathogen-related proteins, such as GmPR10. The way these metabolic pathways are regulated and interconnected remains unclear, but the integration of the data obtained here provides tools to direct studies that aim to fully elucidate the crosstalk between these tolerance/resistance mechanisms, in addition to favoring the development of new soybean cultivars less susceptible to nematodes.

## Figures and Tables

**Figure 1 plants-11-02744-f001:**
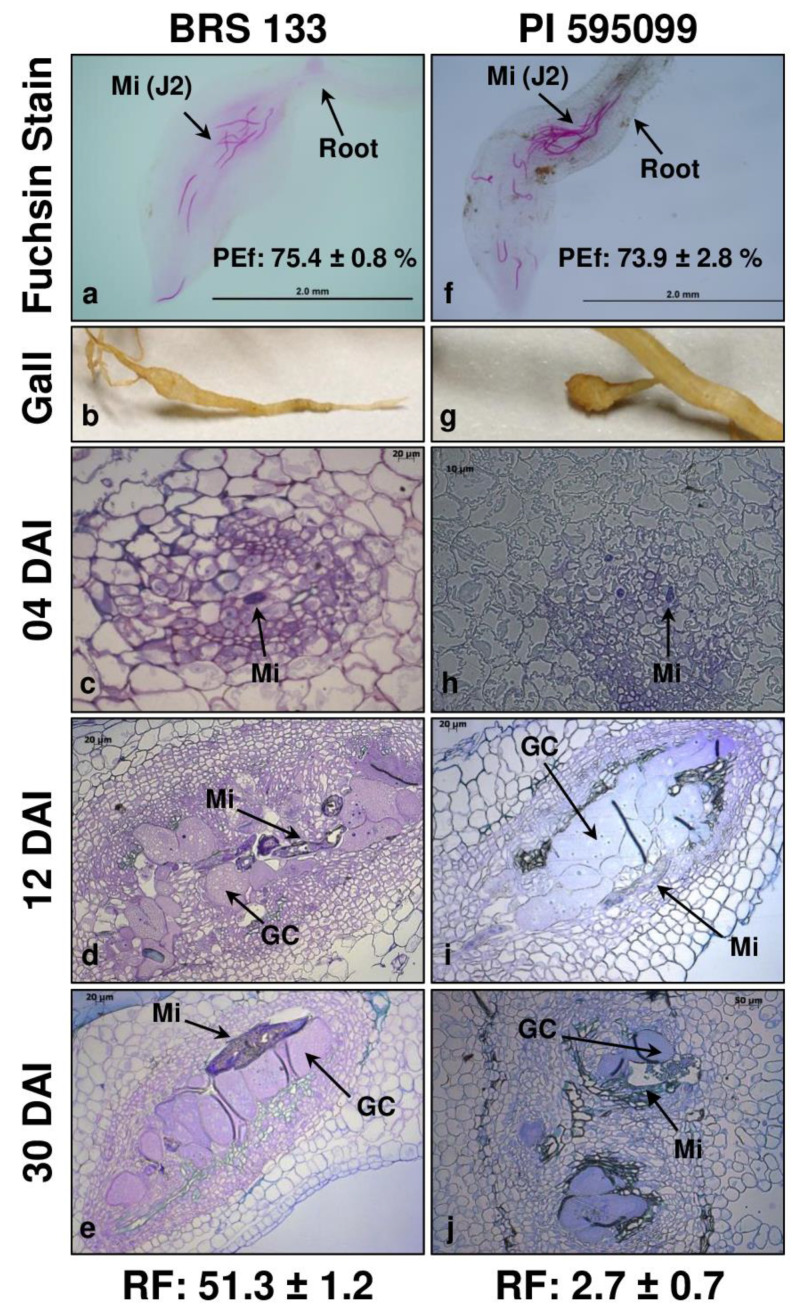
Morphological analysis showing the interaction of soybean roots from two contrasting soybean genotypes: BRS 133 (**a**–**e**) and PI 595099 (**f**–**j**), inoculated with the nematode *M. incognita*. In (**a**,**f**), root tips inoculated with the nematode are presented 4 days after inoculation (DAI). The penetration coefficient (PEf) is similar for both genotypes. In (**b**,**g**), the observed differences in morphology from the galls are shown (BRS 133 is elongated while PI 595099 is ovoid). Comparing the galls or root tips isolated from BRS 133 (**c**–**e**) and PI 595099 (**h**–**j**) evidence that the time course of *M. incognita* infection in both genotypes is similar. Only a slight delay in the development of galls and a smaller cytoplasmic volume in the giant cells were observed in PI 595099 samples. **GC**—giant cell; **J2**—juvenile phase 2 of *M. incognita* development; **Mi**—*M. incognita*; **PEf**—penetration efficiency; **RF**—reproduction factor.

**Figure 2 plants-11-02744-f002:**
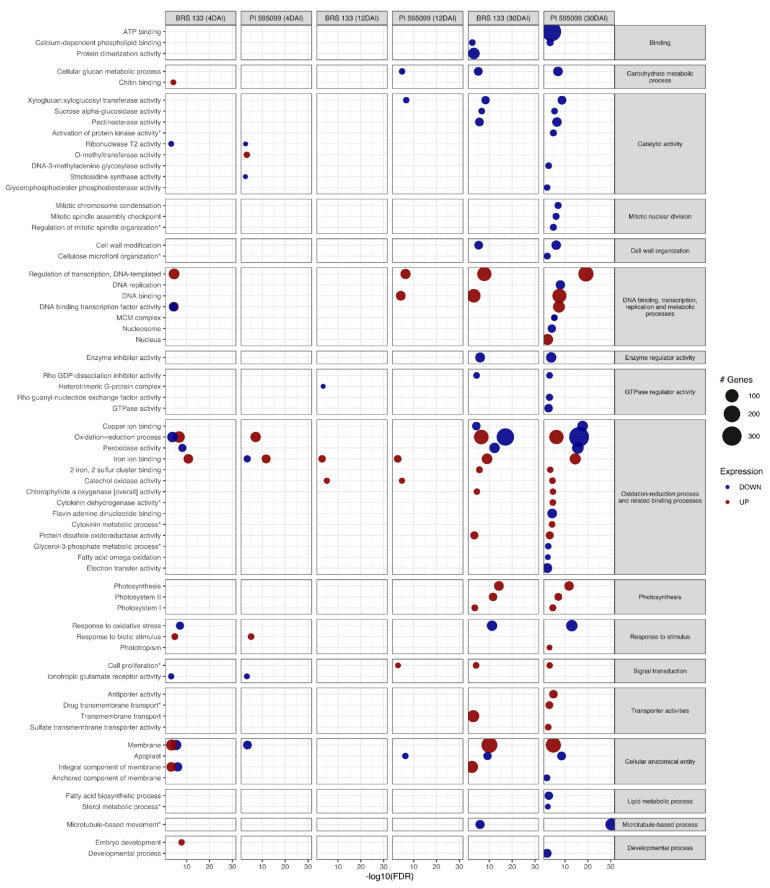
Enrichment of Gene Ontology (GO) terms in the transcriptome of *G. max* infected with *M. incognita*. The susceptible BRS 133 and the tolerant PI 595099 genotypes are represented at the top of the bubble chart, followed by the number of days after the infection (DAI) with *M. incognita*. The GO terms are listed on the left (y-axis). Only GOs with statistical significance (false discovery rate, *FDR* > 0.05) are included.

**Figure 3 plants-11-02744-f003:**
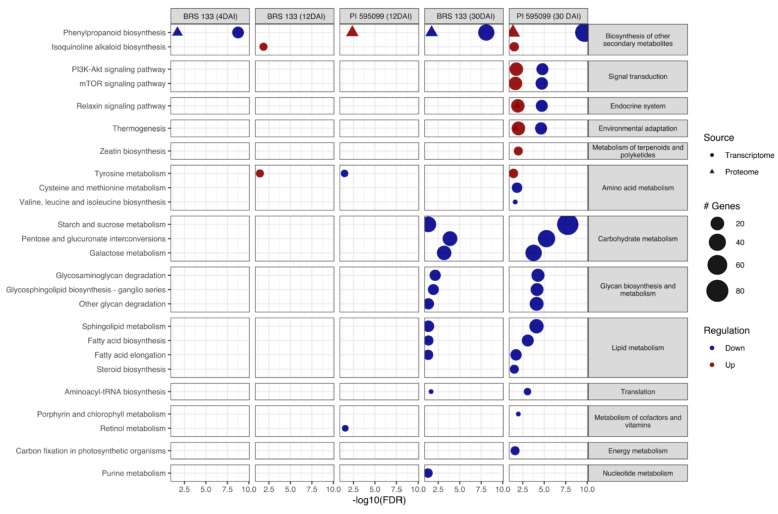
Enrichment of metabolic pathways from Kyoto Encyclopedia of Genes and Genomes (KEGG) database in the transcriptome and proteome of *G. max* infected with *M. incognita*. The susceptible BRS 133 and the tolerant PI 595099 genotypes are represented at the top of the bubble chart, followed by the number of days after the infection (DAI) with *M. incognita*. Major metabolic pathways are listed on the right (y-axis). Minor metabolic pathways are listed on the left (y-axis). Only pathways with statistical significance (false discovery rate, *FDR* > 0.05) are included. No enriched terms were Identified to tolerant PI 595099 genotype at 4 DAI.

**Figure 4 plants-11-02744-f004:**
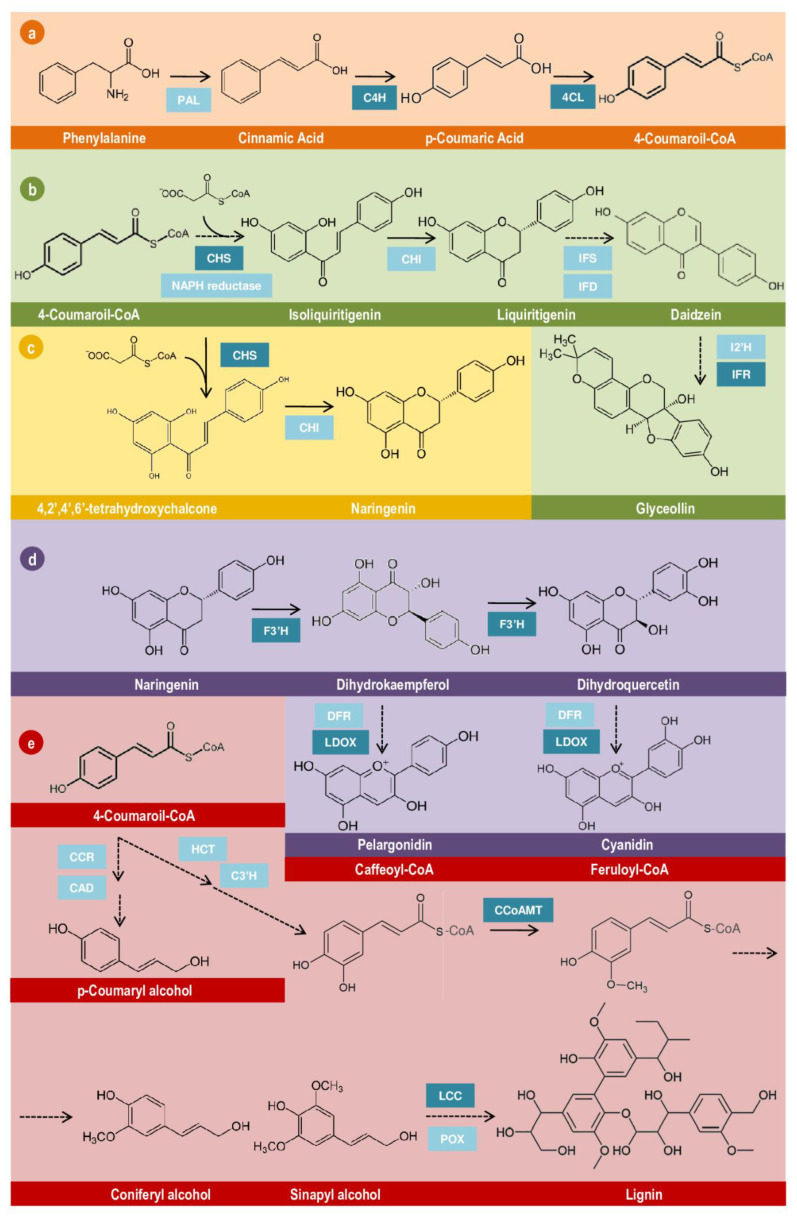
Phenylpropanoid-derived metabolic pathways associated with soybean tolerance to *M. incognita*. According to the proteome and transcriptome analyses, the tolerance to root-knot nematode observed in the soybean genotype PI 595099 might be associated with a super production of secondary metabolites derived from the phenylpropanoids pathway, which directly affect nematodes parasitism. (**a**) Main branch of the phenylpropanoids pathway that converts the aromatic amino acid phenylalanine into p-Coumaric Acid and 4-Coumaroil-CoA. Phenylalanine is converted into cinnamic acid by the enzyme Phenylalanine Ammonia-Lyase (PAL) and subsequently into p-Coumaric Acid by the Cinnamate-4-Hydroxilase (C4H) and upregulated in the tolerant genotype. The p-Coumaric Acid is finally converted into 4-Coumaroil-CoA by the 4-Coumarate-CoA-Ligase, and also upregulated in the highly tolerant genotype. p-Coumaric Acid and 4-Coumaroil-CoA are the molecular backbones for the aromatic-derivative metabolic routes. (**b**) Pterocarpans-derived pathway, converging to glyceollin biosynthesis. The initial and final steps, comprising the reaction catalyzed by Chalcone Synthase (CHS) and Isoflavone Reductase (IFR), are hyperactive in the tolerant genotype. (**c**) Flavanone biosynthesis pathway, converging to naringenin production. The initial step of this derivative route is shared between flavanones and pterocarpans through the reaction catalyzed by the CHS. (**d**) Anthocyanins-derived route. Flavonoid-3′-Hydroxylase and Leucoanthocyanidin Dioxygenase (F3′H and LDOX), the main enzymes involved in the synthesis of the active anthocyanins pelargonidin and cyanidin, are upregulated in the tolerant genotype. (**e**) Lignin biosynthesis relies on the polymerization of aromatic alcohols derived from phenylalanine and tyrosine. A step of this pathway, which converts p-Coumaryl Alcohol into Coniferyl Alcohol and Sinapyl Alcohol (catalyzed by the enzyme Caffeoyl-CoA-3-o-Methyltransferase—CCoAMT) is upregulated in PI 595099 genotype. The final steps of lignin polymerization, catalyzed by cyclic reaction of Laccases (LCC) and Peroxidases (POX), are also upregulated in PI 595099. **Light blue squares:** no changes detected in expression levels. **Dark blue squares:** upregulated enzymes. **4CL**—4-Coumarate-CoA-Ligase; **C3′H**—p-Coumaroyl-5-O-Shikimate 3′-Hydroxylase; **C4H**—Cinnamate-4-Hydroxilase; **CAD**—Cinnamyl Alcohol Dehydrogenase; **CCoAMT**—Caffeoyl-CoA-o-Methyltransferase; **CCR**—Cinnamoyl-CoA Reductase; **CHI**—Chalcone Isomerase; **CHS**—Chalcone Synthase; **DFR**—Dihydroflavonol 4-Reductase; **F3′H**—Flavonoid 3′-Hydroxylase; **HCT**—Hydroxycinnamoyl-CoA: Shikimate Hydroxycinnamoyl Transferase; **I2′H**—Isoflavone-2′-Hydroxylase; **IFD**—2-Hydroxy-Isoflavone Dehydrase; **IFR**—Isoflavone Reductase; **IFS**—2-Hydroxy-Isoflavone Synthase; **LCC**—Laccase; **LDOX**—Leucoanthocyanidin Dioxygenase; **PAL**—Phenylalanine Ammonia-Lyase; **POX**—Peroxidase.

**Figure 5 plants-11-02744-f005:**
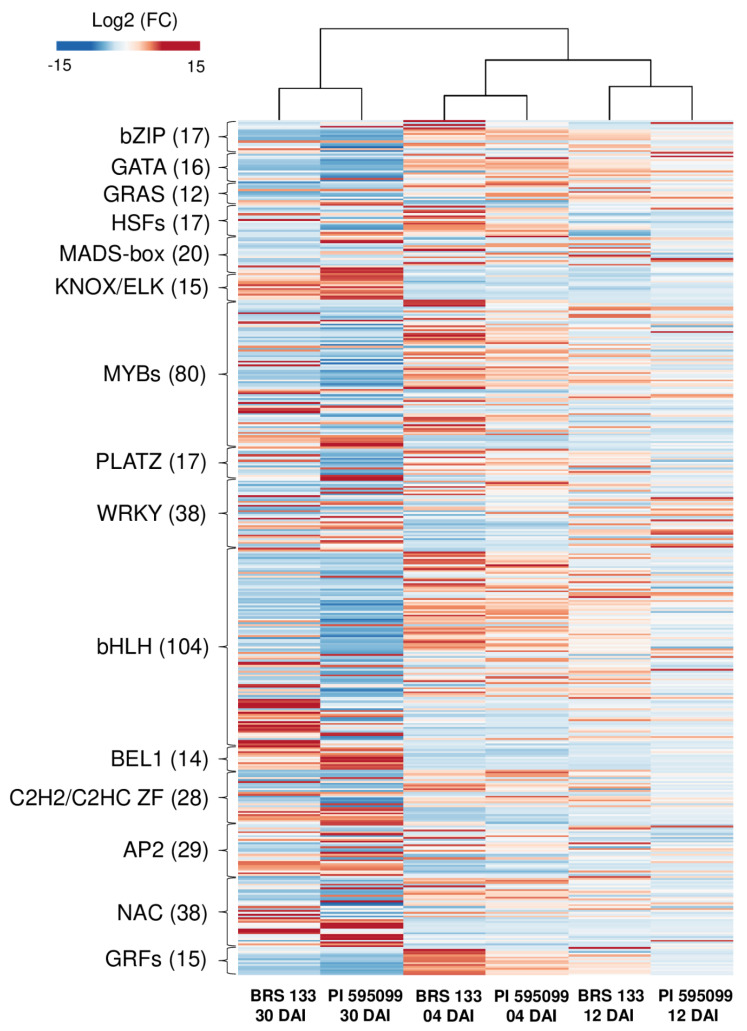
Distribution of differentially expressed genes encoding transcription factors (TFs) in the transcriptomes of genotypes BRS 133 and PI 595099 infected with *M. incognita*. In the heatmap are clusters of the following families of TFs: basic leucine zipper domain (bZIP), GATA family (GATA), GRAS-domain (GRAS), heat shock factors (HSFs), DNA-binding MADS domain (MADS-box), KNOX/ELK homeobox (KNOX/ELK), DNA-binding domain MYB domain (MYBs), plant AT-rich sequence- and zinc-binding (PLATZ), DNA-binding domain WRKY domain (WRKY), basic helix-loop-helix TFs (bHLH), homeobox protein BEL1 (BEL1), c2h2/c2hc zinc fingers (C2H2/C2HC ZF), apetala 2 (AP2), DNA-binding domain NAC domain (NAC) and growth regulating factors (GRFs). Numbers shown in parentheses after the TF families correspond to the number of DEGs. The susceptible genotype BRS 133 and the highly tolerant genotype PI 595099 are represented at the bottom of the chart. Column clusters were generated based on Euclidean distances. Values of fold-change (FC) for each genotype are given in comparison to the time zero [0 days after inoculation (DAI) with *M. incognita*] of the corresponding genotype.

**Figure 6 plants-11-02744-f006:**
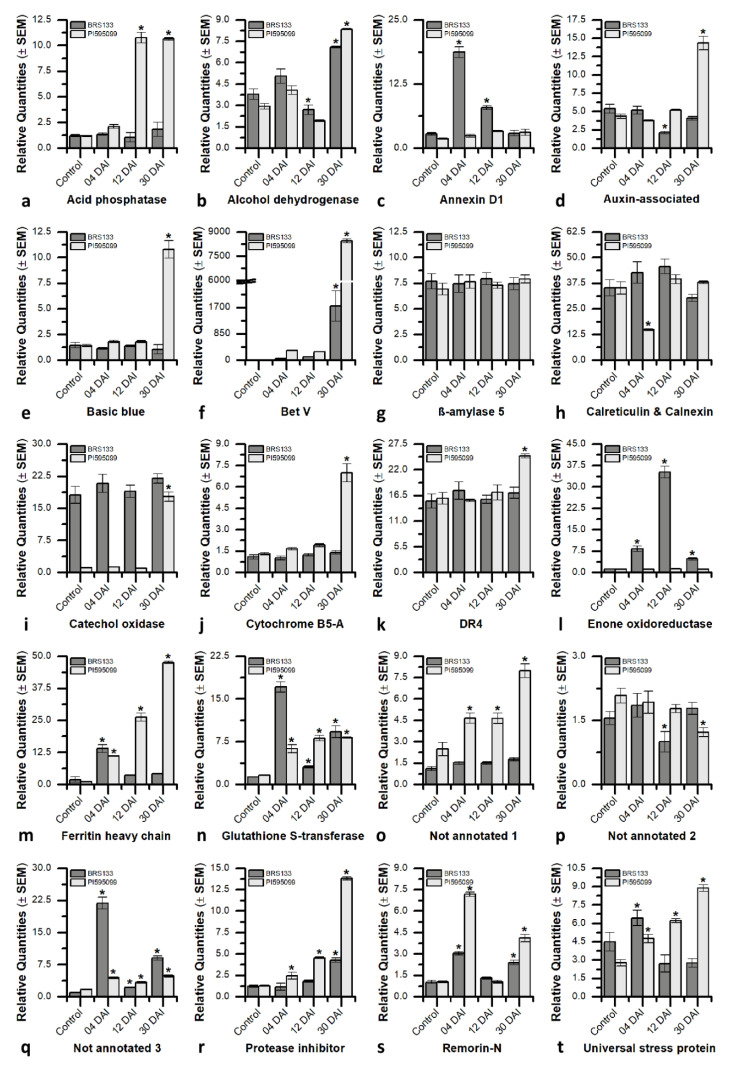
Validation of differentially expressed genes profile. The graphs show the RT–qPCR relative expression (2^ΔΔCt^) of 20 candidate genes selected after comparison of transcriptome and proteome data from two soybean genotypes (BRS 133 and PI 595099) inoculated (4, 12, and 30 days after inoculation—DAI) and not inoculated (0 DAI) with the nematode *M. incognita*. (**a**) acid phosphatase-related (*GmAPHO*; Glyma.08G200100.1); (**b**) alcohol dehydrogenase related (*GmALCD*; Glyma.04G240800.1); (**c**) annexin D1-related (*GmAND1*; Glyma.13G088700.1); (**d**) auxin associated protein (*GmAUXA*; Glyma.13G237000.1); (**e**) basic blue protein (*GmBaBl*; Glyma.08G128100.1); (**f**) pathogenesis-related protein Bet v-1 family (*GmBetV* or *GmPR10*; Glyma.17G030400.1); (**g**) β-amylase 5-related (*GmBAM5*; Glyma.06G301500.1); (**h**) calreticulin and calnexin (*GmCALN*; Glyma.10G147600.1); (**i**) catechol oxidase; tyrosinase (*GmTYR*; Glyma.15G071200.1); (**j**) cytochrome B5 isoform A (*GmCYB5*; Glyma.03G259600.1); (**k**) DR4 protein-related (*GmDR4R*; Glyma.09G155500.1); (**l**) 2-methylene-furan-3-one reductase; enone oxidoreductase (*GmENOX*; Glyma.19G008500.1); (**m**) ferritin heavy chain (*GmFTH1*; Glyma.01G124500.1); (**n**) glutathione S-transferase (*GmGST*; Glyma.18G190300.1); (**o**) not annotated 1 (*GmNOA1*; Glyma.01G018000.1); (**p**) not annotated 2 (*GmNOA2*; Glyma.06G056000.1); (**q**) not annotated 3 (*GmNOA3*; Glyma.19G114700.1); (**r**) trypsin and protease inhibitor (*GmTRPI*; Glyma.15G211500.1); (**s**) remorin, N-terminal region (*GmREMN*; Glyma.09G139200.1); and (**t**) universal stress protein family (*GmUSP*; Glyma.04G107900.1). The asterisks represent statistical significance in relation to the control sample (not inoculated) (*t* test, Bonferroni corrected; *p* ≤ 0.05).

**Figure 7 plants-11-02744-f007:**
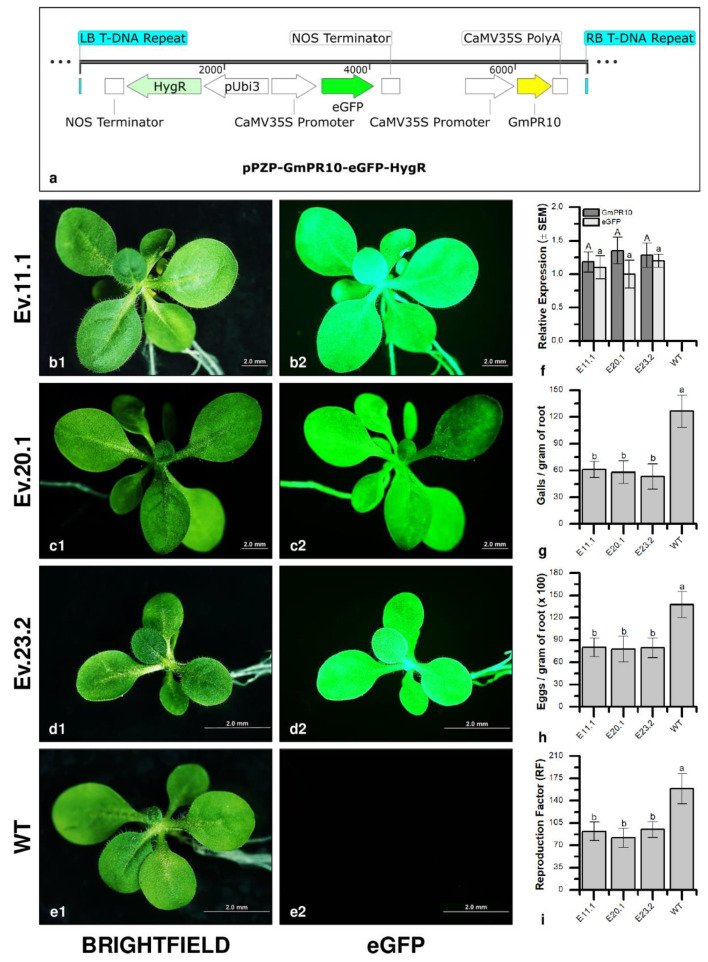
*GmPR10* overexpression in transgenic *N. tabacum*. (**a**) schematic representation of the transformation cassette cloned into the pPZP vector. The *GmPR10* and *eGFP* (a selective marker) genes are both expressed through the *CaMV35S* promoter. Three independent transformation events were selected based on hygromycin resistance and eGFP fluorescence at 488 nm. (**b1**–**e1**) bright-field images of Ev11.1, Ev20.1, Ev23.2, and non-transformed wild-type (WT) tobacco plants and (**b2**–**e1**) images of eGFP fluorescence in the same plants. The *GmPR10* and *eGFP* overexpression was evaluated by RT–qPCR as observed in (**f**). At 60 days after inoculation (DAI), the following parameters were analyzed in soybean-*M. incognita* bioassays: (**g**) galls per gram of root; (**h**) eggs per gram of root; and (**i**) *M. incognita* reproduction factor. The letters A/a/b presented in (**f**–**i**) correspond to groups of ANOVA statistical analysis.

**Figure 8 plants-11-02744-f008:**
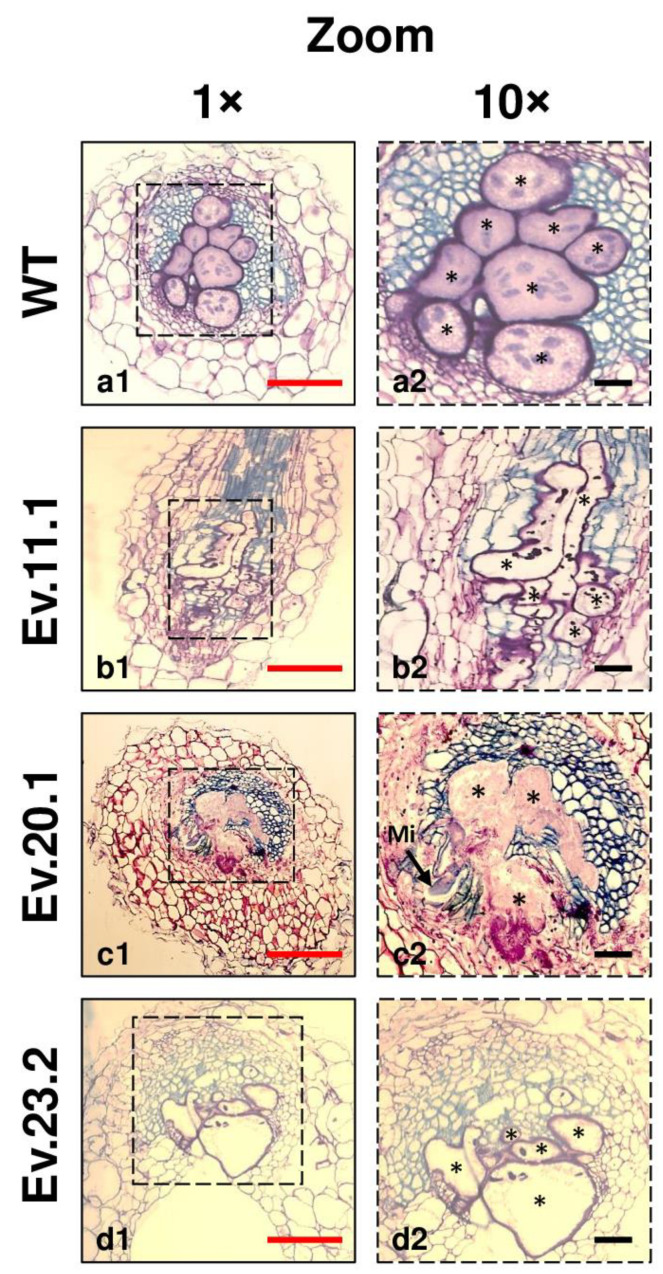
Histological analysis of *M. incognita*-induced galls in transgenic *Nicotiana tabacum* overexpressing the soybean *GmPR10* gene. At least five biological replicates from each line were infected with *M. incognita,* and gall and nematode morphology were evaluated 60 days after inoculation (DAI). (**a1**,**a2**) non-transformed wild-type (WT); (**b1**,**b2**) E11.1, (**c1**,**c2**) E20.1 and E23.2 (**d1**,**d2**). All sectioned galls were stained with toluidine blue. Gall from WT control presented multiple giant cells filled with dense cytoplasm and contained large nuclei. In contrast, galls from transgenic lines showed giant cells with few cytoplasm contents in E11.1 and E23.2 lines and, additionally, E20.1 line showed nematode with altered morphology and giant cells, which presented thinner cell walls. In this way, the analysis demonstrated that *GmPR10* overexpression showed a direct effect on feeding site ontogeny. (**a1**–**d1**) zoom 1×. (**a1**–**d2**) zoom 10×. Legend: (*) *giant cell*; (Mi) *M.*
*incognita.* Scale bars: 50 µm (red) and 2.5 µm (black).

**Table 1 plants-11-02744-t001:** Summary of paired-end reads produced by the sequencing of cDNA libraries from two soybean genotypes.

**Parameters**	**Raw Data**							
	**BRS 133**				**PI 595099**			
	**00 DAI**	**04 DAI**	**12 DAI**	**30 DAI**	**00 DAI**	**04 DAI**	**12 DAI**	**30 DAI**
Reads sequenced (10^6^ reads)	185.9	194.5	202.4	404.0	185.5	182.2	200.3	364.1
Read length (bases)	35–150	35–150	35–150	35–150	35–150	35–150	35–150	35–150
GC content (%)	45.0	45.0	45.0	43.3	45.0	45.0	45.0	44.0
**Total reads sequenced**	**986.832.628**				**932.062.166**			
**Parameters**	**After Clipping and Trimming**							
	**BRS 133**				**PI 595099**			
	**00 DAI**	**04 DAI**	**12 DAI**	**30 DAI**	**00 DAI**	**04 DAI**	**12 DAI**	**30 DAI**
Filtered reads (10^6^ reads)	152.7 (82.2%)	166.1 (85.4%)	172.9 (85.4%)	359.7 (89.0%)	158.4 (84.9%)	154.8 (84.9%)	170.8 (85.3%)	313.9 (86.2%)
Read length (bases)	100–130	100–130	100–130	100–130	100–130	100–130	100–130	100–130
GC content (%)	45.0	44.8	45.0	43.3	45.0	44.8	44.8	44.0
Pseudoaligned reads (10^6^ reads)	140.9 (92.3%)	153.7 (92.6%)	158.9 (91.9%)	276.4 (76.9%)	146.5 (92.5%)	142.8 (92.3%)	157.7 (92.3%)	275.3 (87.7%)
**Total filtered reads**	**851.357.028 (86.3%)**				**797.884.420 (85.6%)**			
**Total pseudoaligned reads**	**729.876.278 (85.7%)**				**722.326.152 (90.5%)**			

**Table 2 plants-11-02744-t002:** Summary of proteome data from BRS 133 and PI 595099 soybean genotypes.

Genotype	Condition	Protein Groups
BRS 133	Control (00 DAI)	2868
	04 DAI	2961
	12 DAI	2872
	30 DAI	2761
**TOTAL**		**11,462**
PI 595099	Control (00 DAI)	2918
	04 DAI	2813
	12 DAI	2991
	30 DAI	2766
**TOTAL**		**11,488**

**Table 3 plants-11-02744-t003:** Elements of phenylpropanoid-derived metabolic pathways differentially expressed in soybean BRS 133 (S) and PI 595099 (T) genotypes.

Enzyme	Abbreviation	Main Pathway	Overexpressed						Underexpressed					
			04 DAI		12 DAI		30 DAI		04 DAI		12 DAI		30 DAI	
			S	T	S	T	S	T	S	T	S	T	S	T
4-Coumarate-CoA-Ligase	4CL	Flavanones		1										1
Caffeoyl-CoA-3-o-Methyltransferase	CCoAOMT	Lignins/Lignans/Phenylpropenes		1		1								
Chalcone and Stilbene Synthase	CHS/STS	Chalcones/Stilbenes						4						
Cinnamate-4-Hydroxylase	C4H	Flavanones					1							
Cinnamyl-Alcohol Dehydrogenase	CAD	Lignins/Lignans/Phenylpropenes												1
Flavonoid 3′-Hydroxilase	F3′H	Anthocyanins/Protoanthocyanidins										1		
Flavonoid 6′-Hydroxilase	F6′H (P450)	Isoflavonols						1						
Isoflavone-4-o-Methyltransferase	IOMT	Resorcinols		1				1						
Isoflavone Reductase	IFR	Glyceollins												1
Laccase ^1^	LCC	Lignin		1										12
Leucoanthocyanidin Dioxygenase	LDOX	Anthocyanins		1			1							

^1^ Different laccase isoenzymes were grouped into a single field.

## Data Availability

The datasets supporting the conclusions of this article are available in the NCBI SRA repository under Bioproject PRJNA750661 (https://www.ncbi.nlm.nih.gov/sra, accessed on 21 July 2021) (Deposited Biosamples with their corresponding cDNA libraries are identified from SAMN20475803 to SAMN20475810). The proteomic data are available in ProteomeXchange and were identified by the ID PXD028483.

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
