# Peer review of "Integrated Omic Approaches Reveal Molecular Mechanisms of Tolerance during Soybean and Meloidogyne incognita Interactions"

_plants, 2022, doi:10.3390/plants11202744_

Round 1
Reviewer 1 Report
General
The authors investigate the proteome and transcriptome of contrasting Glycine max genotypes during interaction with Meloidogyne incognita, an important root-knot nematode. The manuscript is generally well written and well presented. The data presented are relevant for engineering resistance to RKN in soybean.
With regard to the title, the authors should check whether the focus is actually on both tolerance and susceptibility, or whether mostly on tolerance – I suggest correcting the title
It is not immediately clear as to why was GmPR10 selected for overexpression, rather than another candidate gene – the authors need to justify this more clearly in the paper – perhaps early on in the Abstract and Objectives (ie. based on the high expression observed amongst candidate genes)
Discussion of contrasting results with transcriptomic and proteomic data should be further recognized and discussed
All datasets (with and without differential expression) should be available as supplementary files – the reader will expect to have access to the gene data.
Abstract
Line 32. Please make clear to the reader exactly which timepoints of the interaction are being investigated, whether they are pre-penetration or post-penetration.
Line 35. When highlighting constitutive responses to the nematode, the authors should clarify here whether these represent common responses across both the resistant and susceptible genotypes
Line 37. Prior to focusing on the phenylpropanoid pathway, a broader description of the gene expression changes observed (GO, KEGG, TFs) should also be mentioned – and highlight if the changes were exclusive to the highly tolerant PI 595099.
Line 39. There needs to be an explanation as to why PR10 was selected for validation in planta rather than other phenylpropanoid pathway candidates
Line 42. Unclear as to what synchronized metabolic pathways means here
Introduction
Line 51. Delete …overseas
Line 59. correct to …and represents
Line 75. Correct allowed to enabled
Line 79. Correct to …the co-evolution and selection of virulent nematode populations has now been reported….
Line 81. Correct to…As an alternative, the integration of omics can potentially uncover new sources….for exploration by either…
Line 89. Correct to…while BRS 133, by contrast, …..
Line 95. Correct to…Regardless of this,
Background information in the Introduction regarding the state of the art of legume-RKN plant immune responses (in terms of PTI and ETI) and indeed an overview of key previous soybean-RKN data would be useful to contextualize the findings presented in the paper
Materials and Methods
Line 587. Correct to: The genotypes selected for this study comprised BRS 133,
Line 600. Correct to: At 60 DAI (representing two nematode life cycles)….inoculated roots, also according to [112], then later counted….
Line 605. Correct to: A second bioassay was conducted using the same methodology as that employed for RF determination, for collection of biological material for use in transcriptome….
Line 607. Correct to:…where samples of inoculated roots….., as well as from non-inoculated….
Line 610. Please re-write this sentence – it is poorly written and unclear with regard to the differences at 4 and 12 and 30 DAI
Line 619….as described above ??
Line 620…one non-transformed…three transgenic
Line 658..and for subsequent RT-qPCR analysis
Line 666. Lastly
Results
Line 162. Given the activation of phenylpropanoid metabolism, microscopy confirming the accumulation of polyphenolics in the root sections would be useful, if available
Line 179. The authors should make available their DEG and DEP data – listing all genes and proteins expressed and differentially expressed. Thes should ideally be as supplementary excel files
Line 197. Do the authors (in the Discussion) hypothesize as to why DEGs and DEPs were so different, with only 4% or 1% common across evaluated time points ?
Line 301. With the authors focusing mainly on phenylpropanoids and TF expression, PR10 becomes an unexpected choice for validation in planta. I suggest the authors provide more information on the abundance of PR DEGs in the datasets, to justify further the selection of GmPR10.
Table S3 – Include a column with the gene description information. Please clarify that all differentially expressed genes are statistically significant, in terms of FDR.
Discussion
Line 346. Please highlight whether in reference 44 the authors focused on transcriptional profiling of a resistant, tolerant or susceptible genotype of soybean – to know whether the previously observed genes involved in pathogenesis, cell cycle and cell wall represent resistance responses or not
Line 367. The authors need to explain possible reasons for the often-contrasting results observed with proteomic v transcriptomic analyses
General – the authors should also describe in the paper (Results and Discussion) whether other key elements of the immune response were observed in the datasets – for example originating from PTI or ETI tiers of the immune system.
Reviewer 2 Report
The manuscript “Integrated omic approaches reveal molecular mechanisms of tolerance/susceptibility during soybean and Meloidogyne incognita interactions” by Arraes et al. compares the responses of susceptible (BRS 133) and tolerant (PI 595099) soybean genotypes during interaction with the root-knot nematode Meloidogyne incognita at different stages of infection. The authors carried out transcriptome and proteome analyses as well as microscopic study of galls and giant cells. A special focus was put on the synchronized overexpression of soybean PR10 genes and genes associated with phenylpropanoid biosynthesis that could improve the protective effect against infestation by M. incognita.
The manuscript is well written, the Material and method section is well organized, the presentation of the results is mostly good. There are, however, some points that need clarification:
Results
Lines 336-343 Given that the histological and transcriptomic data on soybeans were acquired up to 30 DAI, why were the morphological studies on transgenic tobacco lines carried out at 60 DAI? I think that an analysis at this time point is risky because the infection found is by second-generation nematodes. This implies that you can find galls of different ages that would also justify the different appearance observed in the images. How many galls per transgenic line have been observed? and how many sections? Why some sections are transversal and others longitudinal? In this way, it is difficult to make comparisons.
In addition, it is not possible to establish the nematode in transgenic roots had altered morphology if there is not the possibility to compare its appearance in a WT gall. Moreover, in figure 8c2 only a small portion of the nematode is visible and altered morphology is not appreciable. I suggest to include additional pictures showing nematode morphology.
Figure 1 Authors state “Only a slight delay in the development of galls, as well as a smaller cytoplasmic volume in the giant cells” is appreciable in PI 595099 samples. No mention on how authors calculated giant cell volume is present in the manuscript. If differences have been deduced only by using different magnifications (i.e. for the tolerant genotype), I recommend to better specify it in the caption or highlight the magnification bar.
Material and Methods
M. incognita race 3 population has been used in experiments with transgenic lines of Tobacco. According to the NC differential host test (Hartmann and Sasser, 1985) M. incognita race 3 populations don’t reproduce on Tobacco. How can you explain the capability of M. incognita to infect WT tobacco plants and even if to a lesser extent the transgenic lines?
Minor points:
Introduction
Line 65 Please change “from the pharingeal gland” in “the esophageal gland cells” as they are three large specialized secretory gland cells, one dorsal and two subventral.
Results
Line 201 Do you mean table S3? Please check.
Line 307 According to the graph (figure 6f) it seems that the expression level at 30 DAI is higher than 5,500 times respect to the non-inoculated control. Please check.
Line 309 Remove a dot after the word conditions.
Line 313 Please change “04-09” in “4-9”
Line 335 “Overall, transgenic plants exhibited a reduction of 40.0-58.0% in M. incognita reproduction.” How did you calculate this reduction?
Figure 3 Enrichment by KEGG of the tolerant PI 595099 genotype at 4 DAI is missing. Is it correct? If this depends on the fact that there is no enrichment for any metabolic pathway for this dataset it is appropriate to specify in the results
Figure 4 It is not clear to me the meaning of green arrows. Please explain.
Figure 8 In the caption are mentioned “gall 1 and gall 2” that are not shown in the figure.
Discussion
Line 539 The term “protective” sounds inappropriate as the most commonly reported biological function of PR10/Bet v1-like proteins is anti-pathogen activity against viruses, bacteria, fungi or nematodes and in this sentence you are describing a negative effect of PR10 on nematodes.
Material and Methods
Lines 595-599 I suggest shortening this sentence, i.e. “each soybean plant was inoculated with 1,350 J2 of M. incognita race 1 population obtained from eggs collected from a culture maintained on tomato plants (Solanum lycopersicum L. cv. Santa Clara) in greenhouse”.
Line 608 roots “at” 4, 12 and 30 DAI
Line 618 I find reference 113 inappropriate
Lines 619 and 644 Please change “below” with “above”
Line 708 Please correct Trizma base
Lines 801-802 Clear the dots in μg.mL-1
Conclusions
Line 868 Please change NOS in NO
